# Droughts and Media: when and what do the newspapers talk about the droughts in England?

Inhye Kong[1], Jan Seibert[1], Ross S. Purves[1]

[1]Department of Geography, University of Zurich, Zurich 8057, Switzerland

*Correspondence to*: Inhye Kong (inhye.kong@geo.uzh.ch)

**Abstract.** The United Kingdom is traditionally known for its wet climate, but droughts are also a recurring concern. Using newspapers as a medium of public communication, this study explores the timing and content of newspaper articles about droughts in England. We constructed a corpus of more than 800 newspaper articles related to droughts in England for the last 24 years (2000 - 2023) and analysed the temporal alignment of newspaper coverage with hydroclimatic anomalies and

seasonality using a negative binomial regression model. Our result showed that newspaper coverage of droughts coincides with short-term shortage of precipitation (SPI-3) and groundwater (SGI-1) in spring/summer seasons, but temperature (CET-12) was not a significant factor. Using topic modelling we explored narratives found in the texts such as "Drought and hosepipe ban", a common measure to restrict water usage in England during periods of hydrological drought. Comparing two major droughts (i.e. spring 2012 vs. summer 2022), we found the summer drought of 2022 to contain more summer-related topics

(i.e., "Heatwave", "Temperature and hosepipe ban"), which underpins our statistical analysis suggesting that warm seasons garner more media attention. Overall, our findings reveal that newspaper reporting about droughts is influenced by a combination of factors, rather than precipitation alone, with a notable seasonality-based component which may reflect a confirmation bias in reporting of droughts. This in turn implies a potential mismatch in how droughts are conceptualized by newspapers compared to scientists and may underrepresent early signs of drought in cold seasons, potentially undermining

public support for preventive measures at these times of years.

## 1 Introduction

The United Kingdom (UK) is well known for its wet climate and frequent flooding, but droughts have commonly recurred throughout history, with a major drought every 5-10 years on average (Dayrell et al., 2022; Marsh et al., 2007; Murphy et al., 2006). Within the last two decades, The National Hydrological Monitoring Programme of the UK has released major

drought reports for 2003 (Marsh, 2004), 2004-2006 (Marsh, 2007), 2010-12 (Marsh et al., 2013), and 2018-19 (Turner et al., 2021). England is more susceptible to severe droughts within the UK due to overall less precipitation and greater water demand from human influences (Tanguy et al., 2021; Tijdeman et al., 2018) than other constituent countries. This pattern is especially distinctive around southern England which includes the London metropolitan area (Dessai & Sims, 2010; Folland et al., 2015). According to the most recent IPCC report and other sources (IPCC, 2023; Guerreiro et al., 2018; Spinoni et al., 2018), climate

change is expected to heighten drought-prone conditions in Western and Central Europe, and England is no exception (Dobson et al., 2020; Kay et al., 2023).

Droughts, as an example of extreme weather events, are defined as prolonged periods of imbalance between rainfall and evapotranspiration (Wilhite & Glantz, 1985). They can be further categorised as meteorological droughts, characterised by a lack of precipitation, and hydrological droughts, marked by reduced streamflow, reservoir levels, lake volumes, and

groundwater table (Van Loon, 2015). Aside from these hydroclimatic droughts, socioeconomic droughts are defined by the impact of drought on society, including agricultural and economic losses and failures to meet water demands for people and the environment (Van Loon, 2015). Compared to other natural hazards, droughts appear to receive less media and public attention (Brimicombe, 2022). This is primarily due to their being a "creeping phenomenon" – a gradual change, often developing over extended periods, in contrast to the rapid onset of events like floods and wildfires (Kim et al., 2019). In

addition, droughts are the result of complex hydrological dynamics involving a lagging effect, resulting in temporal mismatches between rainfall, streamflow, and groundwater recharge (Van Loon, 2015): deficits in rainfall (i.e., meteorological drought) do not always coincide with or lead to low streamflow or low groundwater (i.e., hydrological drought), and not all such cases result in tangible impacts for society (i.e. socioeconomic droughts).

Given the complex and gradual nature of drought dynamics, it is crucial to analyse how and when droughts are

reported in the media, as these factors play a significant role in shaping public perception and response. The timing of media coverage reflects not just the occurrence of droughts but also the point at which their impacts become newsworthy (Caple, 2018). This raises important questions about whether media reporting aligns with the actual progression of drought events. Additionally, the framing and narratives constructed in media coverage deserve examination (Vargas Molina et al., 2022), as they influence public perception and societal responses, ranging from individual behavioural changes (Antwi et al., 2022;

Quesnel & Ajami, 2017) to responsiveness to policy initiatives (Hart et al., 2015). Thus, understanding the temporal and narrative dimensions of drought reporting is essential for enhancing public engagement, fostering informed decision-making, and driving effective societal action.

Despite the rise of new and diverse media forms, most prominently social media, newspapers remain a vital data source for media analysis (Vargas Molina et al., 2022). Written by professional journalists and reporters, newspapers often

have large circulations reaching broad and diverse segments of the population (Boykoff, 2008). This enables newspapers to represent a wide range of societal perspectives, which are often absent in social media or scientific publications. Another advantage of newspapers lies in their coverage of diverse topics and issues, reflecting the multifaceted concerns and dynamics of society (Schmidt et al., 2013). Moreover, advancements in digital transformation have greatly improved access to media archives, providing researchers with extensive historical and contemporary newspaper datasets. These developments facilitate

the creation of large-scale newspaper corpora spanning significant temporal ranges, enabling longitudinal analyses (Dayrell et al., 2022; O'Connor et al., 2022).

Leveraging access to extensive digital archives of news media, several studies have investigated the temporal alignment between media attention about droughts and hydroclimatic conditions. A common approach involves using the size

of the news corpus as a proxy for media attention and comparing it with meteorological and hydrological indices. For example, Linés et al. (2017) used drought-related indices, including the Standardized Precipitation Index (SPI), to assess temporal alignments with media reports in Spain. Noone et al. (2017) analysed an Irish newspaper corpus and correlated it with SPI-12 over the past 250 years. Dayrell et al. (2022) examined UK newspaper corpora over the last 200 years to explore the correlation between a drought index (i.e., SPI) and media coverage. O'Connor et al. (2022) studied historical newspaper articles (1900–2016) in Ireland, finding SPI-3 to be the best predictor of land-based drought reports. However, there are some missing factors that need to be considered in understanding the perception of droughts by media. For example, the role of temperature remains underexplored, although Pianta and Sisco (2020) noted that short-term temperature anomalies influence media coverage of climate change. Furthermore, confirmation bias on seasonality in media reporting, as shown in more attention on warmer months (O'Connor et al., 2022; Parsons et al., 2019), needs further investigation to understand the media dynamics of drought reporting.

Regarding content, several studies have examined newspaper articles about droughts and identified dominant topics, main stakeholders, impacts, and emerging concerns. Dow (2010) utilized qualitative content analysis to analyse decades-long news coverage (1998-2007) in the Carolinas US identifying the impacts of droughts (e.g., agriculture, fires, lawns, recreation, etc.). Wei et al. (2015) used qualitative content analysis to explore historical news media reporting of water issues in Australia (1843-2011), revealing institutions and policy initiatives related to different types of water-related events. Osaka et al. (2020) focused on revealing difference in content between nationwide and regional newspapers about Californian droughts between 2011-2017. Painter et al. (2021) collected online news reports of the 2019 summer heatwave across France, Germany, the Netherlands, and the UK, and analysed how climate change was represented in the content. However, most studies rely on a qualitative approach, which poses significant constraints in the scalable application of content analysis. In recent years, natural language processing (NLP) has been applied to process a large text data sets and extract key narratives, for example, Bohr (2020) used unsupervised text analysis with topic modelling about newspapers discussing climate change in the United States, and Sodoge et al. (2023) applied pre-trained text classifiers to classify drought impacts from German newspapers (2000-2021).

The first quarter of the 21st century was the period when broad scientific consensus on anthropogenic climate change was reached, and also a period with more frequent, intense, and widespread drought impacts. UK government policies have clarified a need to introduce programs for climate change adaptation, including relevant programs for droughts (e.g., The Third National Adaptation Programme (NAP3) and the Fourth Strategy for Climate Adaptation Reporting). Still, there is limited evidence as to how people perceive droughts, especially in generally humid regions such as England. The research gap addressed in this paper addresses our limited understanding of the link between the volume of newspaper articles and hydroclimatic conditions. Additionally, there is a need for investigation into newspaper narratives surrounding the coverage of droughts, utilizing scalable computational approaches to enhance our understanding and linking these to drought indices.

In this study, we analysed newspaper articles reporting droughts in England, aiming to 1) quantify the influence of hydroclimatic conditions, including not only precipitation but also temperature and seasonality, on the volume of newspaper articles, and 2) examine the content of newspaper narratives. We hypothesised that newspaper reporting is driven by the

conditions of meteorological droughts (i.e., higher temperatures and lower precipitation) and hydrological droughts (i.e. lower groundwater level), alongside the possibility of confirmation bias in seasonality – that is to say, an expectation that drought-related media coverage is more relevant in warmer seasons. In our content analysis, we expect to find topics related to the different types of droughts (i.e. meteorological, hydrological, and socioeconomic droughts), and to explore distinctions in topics depending on different background conditions. Therefore, our research questions are summarised as follows:

RQ1. How and to what extent do hydroclimatic conditions and seasonality relate to newspaper reporting in England?

RQ2. What are the dominant narratives in newspaper reporting of droughts, and how do these narratives vary across major drought events?

## 2 Methodology

To answer the research questions, we build a newspaper corpus about drought reports in England and analyse the corpora a) to compare the size of media coverage to drought-related anomaly indices and seasonality (RQ1), and to explore media content using natural language processing (RQ2). All the code used in our analysis is available as R-Markdown (see Supplementary Material 1).

### 2.1 Data collection

#### 2.1.1 Corpus building

The first step in any text-related project is creating a systematic corpus of relevant documents. Among many approaches to building a corpus, we chose a keyword-based approach to collect newspaper articles about drought-related reports in England. We chose only England, as opposed to the whole UK, since England's drier climate and higher population density leave it more susceptible to droughts and their impacts (Folland et al., 2015). Besides, current water management in the UK is delegated to the devolved governments of Northern Ireland, Wales and Scotland (Robins et al., 2017), and we aimed to minimise the influence of such regulatory differences.

To collect newspaper articles, we used the media hub platform Nexis Uni (www.advance.lexis.com). Initially, we retrieved articles that contained the term 'drought' in their headline field and were published between January 2000 and August 2023 (i.e. August marks the end of one hydrological cycle). However, since the term drought may be used metaphorically (e.g., 'goal drought' in sports), we further refined the search results to retain only those containing the terms 'dry' and 'England' so that we can maximise both precision (i.e., the proportion of articles retrieved whose topic was related to droughts) and recall (i.e., the proportion of all articles describing droughts in some form that we retrieved).

In making a query, we also defined the source of newspapers. We considered both broadsheet and tabloid newspapers to include a wide-spanning political spectrum (i.e., right- and left-wing stances) as well as readership (i.e., both widely-read tabloid journalism and broadsheets, referred to as the quality press) (Boykoff, 2008; Norton & Hulme, 2019). Our corpus was

based on five broadsheet sources (i.e. The Guardian, The Independent, The Times, Financial Times, and The Daily Telegraph)

and three tabloid sources (i.e. The Sun, Daily Mirror, Daily Mail). When the source of newspapers was not identified, the articles were removed. In the end, we obtained 1,361 drought-related articles.

Having built the corpus, we carried out data-cleaning steps to increase precision (i.e., the proportion of relevant articles from the queried results). We first removed duplicate articles with the same title. Then, we screened the geographic and thematic metadata provided by Nexis Uni and removed those where primary entries were irrelevant (e.g. geographic

metadata referring to Wales, Scotland, Northern Ireland, and abroad; subject metadata referring to sport). Meanwhile, we noticed several cases of similar articles remained in the corpus, where, for example, online articles were very slightly edited versions of printed articles or vice-versa, but they were kept in the corpus. After these steps, 836 articles (61%) remained for the analysis, and we grouped these to calculate monthly counts. The process was performed in R Studio version 4.3.2 (R Core Team, 2023)


### 2.1.2 Hydroclimatic and seasonality metrics

For meteorological and hydrological variables, we considered temperature, precipitation, streamflow, groundwater, and seasonality. For the temperature, we collected Mean Central England Temperature (°C) (hereafter CET; Parker et al. 1992, https://www.metoffice.gov.uk/hadobs/hadcet/data/meantemp_monthly_totals.txt) from the web archive of the UK

Meteorological Office (MET), the United Kingdom's national weather service. To calculate temperature anomalies, we first calculated long-term averages of CET for 1991-2020 (30 years), which served as baselines, and then computed anomalies, i.e., deviations of CET from the long-term baseline. The temperature anomalies were then computed for moving windows of 3, 6, and 12 months (e.g., for an event in January, the window of 3 months includes January and the preceding December and November), to consider the lagging effect in physical processes.

For the precipitation, streamflow, and groundwater, we collected Standardised Precipitation Index (SPI), Standardised Streamflow Index (SSI) and Standardised Groundwater Index (SGI) from the UK Water Resources Portal (https://eip.ceh.ac.uk/hydrology/water-resources/). The SPI is a popular drought index to estimate the probability of precipitation for a given temporal window (e.g., Dayrell et al., 2022; O'Connor et al., 2022; Parsons et al., 2019). SPI and SSI data were provided with the moving windows of 3, 6, 12 months (i.e. SPI-3, SPI-6, SPI-12, SSI-3, SSI-6, and SSI-12). Since

these data were at station level, we aggregated them to make an England-level index. Here, we found a strong correlation between SPI and SSI (ranging from 0.82 to 0.90, Supplementary Material 2), thus, we only considered SPI in the analysis, to minimise the effects of multicollinearity. The SGI data was also at station-level, but only with 1-month window, so we prepared an aggregated SGI-1 of England.

Finally, we defined seasons as autumn (SON, September-October-November), winter (DJF, December-January-

February), spring (MAM, March-April-May), and summer (JJA, June-July-August), marking September (the onset of autumn) to be the beginning of the hydrological cycle in England.

## 2.2 Newspaper reporting in relation to hydroclimatic anomalies and seasonality

### 2.2.1 Statistical regression analysis

To measure the influence of temperature, precipitation, groundwater, and seasonality on the size of newspaper reporting, we used a statistical regression analysis. We carried out a negative binomial regression, which is optimal when the dependent variable is over-dispersed, i.e. the data does not follow normal distribution and are highly skewed, due to e.g. many zero counts (Green, 2021). Negative binomial regression also allows inclusion of categorical variables into the model, which is necessary for considering seasonality. In the regression, we added an additional variable to explain the lagging effect of

media attention by taking the article counts of the preceding month. This lagging effect of media was meant to capture autoregressive patterns, i.e., when the media attention rises, it can generate a spill-over effect in time. As a result, we constructed a model of monthly counts of newspaper articles as a function of CET, SPI, SGI, seasonality, and the lagging effect of media, where CET, SPI, and SGI refer to the anomalies for temperature, precipitation, and groundwater, and seasonality refers to four classes of seasons where autumn is a baseline. In testing model performance, we arranged different

combinations: there are four options for CET, i.e., 3, 6, 12 months of anomalies, or none; for SPI, likewise, there are four options to choose: SPI of 3, 6, 12 months, or none; for SGI, there were two options: SGI of 1 month or none; for seasonality, four categorical values, i.e. SON, DJF, MAM, JJA, were considered altogether or not. The lagging effect of media was included in all combinations. As such, we tested all possible combinations of attributes including rich and parsimonious models (i.e., $4*4*4*2-1 = 63$) to find the best-fitting model using the MASS package in R (Venables & Ripley, 2002). The best-fitting

model was selected based on the lowest AIC (Akaike Information Criterion).

### 2.2.2 Time-series visual plots

We created time-series plots to show how the selected variables from the best-fit model interplay with the monthly count of newspaper articles. The plot is composed of a bar plot to show the count of monthly newspaper articles, along with line plots of 'the most fitting' variables of CET, SPI and SGI from the regression analysis. The colour-blind friendly palette

for this visualisation was take from Wong (2011). Additionally, we coloured the background yellow for the months where the conditions are favourable to droughts, i.e. $CET > 0$, $SPI < 0$, and $SGI < 0$, which helped us to find the overlaps between the volume of newspaper reporting and the favourable hydroclimatic conditions.

The second temporal plot with a heatmap delivers more visually oriented information showing the monthly article counts and selected variables out of CET, SPI, SGI, along with text labels and colour gradients. To reduce outlier effect in

colour saturations, we applied a winsorisation normalization method to replace the colours for extreme outliers with the colours of statistical endpoints of 1% and 99%.

## 2.3 Newspaper narratives with topic modelling

Topic modelling is an unsupervised computational method to identify and group topics through natural language processing. The algorithm groups words found in documents in a corpus according to their probability of belonging to a topic (Blei et al., 2010). Topics are reported as groups of words associated with probabilities, and researchers often label these in a post-hoc process. Topic modelling is a common unsupervised approach to exploring the content in a corpus through so-called distant reading, without recourse to reading individual articles. It is often effective in giving an overview of different themes found in the text (e.g. Zhang et al., 2021).

We carried out topic modelling using nouns in English, chosen for their important role in delivering key semantics about subjects (e.g. who is talking) and objects (e.g. what do they talk about). We used the pre-trained model in spaCy library (Benoit & Matsuo, 2020) to process our texts and extract lemmatized nouns as inputs for topic modelling. Lemmatising reduces words to a common root, allowing us to treat singular and plural forms as the same term. We then ran structural topic modelling (STM) using the R package 'stm' (Roberts et al., 2019), which employs latent Dirichlet allocation (LDA) for topic clustering. Since topic modelling is an unsupervised method, it is necessary to experiment to find the best-fitting number of topics, k, and we used two measures: coherence, which measures how well the terms in a topic are grouped in a document, and exclusivity, which captures how well topics split documents into different groups.

After performing topic modelling, we arranged the topics according to the descending order of topic proportions. For each topic, we retrieved ten keywords with the highest probabilities and ten for the highest frex scoring algorithm (i.e., keyword extraction methods considering both frequent in and exclusive to a topic of interest) to understand the context of topics. After reviewing the keywords, the authors assigned labels that best describe the topics.

Since STM is a mixed membership model, one article can contain more than one topic. For analytic convenience, we assigned one prime topic for each article based on the highest topical probability. Assigning one single topic to each article is helpful in estimating the count and ratio of topics over time. We further extracted entity names and sorted them by frequently mentioned organization names within the sub-corpora.

To take a closer look at the narratives of newspapers, we chose the drought in the first half of 2012 (Drought in spring 2012 hereafter) and the drought in mid-2022 (Drought in summer 2022 hereafter), which were reported as extreme droughts on the one hand by the official reports of the Centre for Ecology & Hydrology (i.e., Major Hydrological Events) and research papers in the last two decades (e.g., the drought of 2012 by Parry et al., 2013; the drought of 2022 by Barker et al., 2024). By making sub-corpora for two droughts, we summarise topical composition using tree map plots (Tennekes, 2023), together with a colour-blind friendly palette 'safe' from 'rcartocolor' in R (Nowosad, 2018). Then we applied close reading (c.f. Dayrell et al., 2022) to understand how the droughts were represented in context, by looking into headlines and body text corresponding to the topics.

## 3 Results

### 3.1 Newspaper reporting trends about droughts

#### 3.1.1 Hydroclimatic anomalies and seasonality effect on newspaper reporting

For the 63 combinations of input variables, we ran the negative binomial regression and compared their AICs. As a result, we found the best AIC rich model (883.87) for the following combination: CET-12, SPI-3, SGI-1, with seasonality considered. A more parsimonious model had a similar AIC (885.06) but only included SPI-12 and seasonality (Table 1). Full lists of the tested combinations and their AIC can be found in Supplementary Material 3. To graphically explore model performance, we plotted the expected number of newspaper articles from the model and compared them with the actual count of newspaper articles by month (Supplementary Material 4).

The results of the negative binomial regression analysis for the rich and parsimonious models are summarized in Table 1 and we explain here how the variables can be interpreted with respect to our dependent variable, the count of media articles for the rich model. Here, CET-12 was positively associated with the count of media articles ($\beta$=0.395, $p$=0.067), indicating a positive but not statistically significant ($p < 0.05$) trend. The coefficient of $\beta$=0.395 indicates that for each unit increase in the CET-12 index, the count of newspaper articles increases by the factor of $e^{0.395}$ or 1.484, resulting in an increase of approximately 48.4%. Conversely, both the SPI-3 ($\beta$=−0.415, $p$<0.001) and the SGI-1 ($\beta$=−0.868, $p$<0.001) were negatively associated with the count of newspaper articles and statistically significant. For each unit decrease in the SPI-3, the count of newspaper articles increases by 51.5% (i.e. ($1/(e^{-0.415} \approx 0.660)$) - 1 = 0.515), while a unit decrease in the SGI-1 corresponds to an increase of 138.2% (i.e. ($1/(e^{-0.868} \approx 0.420)$) - 1 = 1.382).

Seasonal factors also played a substantial role in modelling the monthly count of newspaper articles. The winter season (i.e. DJF, $\beta$=0.693, $p$=0.030) showed a significant positive effect, indicating a 100% increase (i.e. $e^{0.693}$-1 ≈ 1.000) in newspaper articles during this season compared to the baseline, i.e. autumn season. The spring (i.e., MAM, $\beta$=1.065, $p$<0.001) and summer seasons (i.e. JJA, $\beta$=1.108, $p$<0.001) had even stronger positive relationships, with expected increases of 190% (i.e. $e^{1.065}$-1 ≈ 1.901) and 203% (i.e. $e^{1.108}$-1 ≈ 2.028) respectively, suggesting much higher article counts in these periods than others. The lagging effect of media ($\beta$=0.046, $p$<0.001) was positive and significant, demonstrating an autocorrelation effect.

**Table 1** Statistical results of negative binomial regression models: the best AIC model and the parsimonious model (*p*-values: '***' < 0.001, '**' < 0.01, '*' < 0.05, '.' < 0.1)

| Variables | The best AIC model | | Parsimonious, yet 3<sup>rd</sup> best model | |
|---|---|---|---|---|
| | Best-fitting variable | Negative binomial coefficient (p-values) | Best-fitting variable | Negative binomial coefficient (p-values) |
| Temperature | CET-12 | 0.395 (.) | - | - |
| Precipitation | SPI-3 | -0.415 (***) | SPI-12 | -0.835 (***) |

| | | | | | | |
|---|---|---|---|---|---|---|
| Groundwater | SGI-1 | | -0.868 (***) | | - | - |
| Seasonality (Autumn as baseline) | To include | Winter | 0.693 (*) | To include | Winter | 0.670 (*) |
| | | Spring | 1.065 (***) | | Spring | 1.107 (***) |
| | | Summer | 1.108 (***) | | Summer | 1.206 (***) |
| Lagging effect of media attention | To include | | 0.046 (***) | To include | | 0.050 (***) |
| Intercept | | | -0.736 (**) | | | -0.528 (*) |

### 3.1.2 Temporal dynamics between hydroclimatic anomalies and newspaper reporting

The combined plot of the volume of newspaper articles and hydroclimatic anomalies (Figure 1) shows how the

255 reporting of newspaper co-developed with drought-prone conditions. From this plot, we found multiple drought events gathered substantial newspaper reporting, aligning with major drought events declared in governmental reports and scientific articles, e.g., in 2005-2006; 2011; 2017 and 2018/19 (Turner et al., 2021) and most markedly in 2022 (Barker et al., 2024). The plot also showed that most newspaper reporting for major droughts coincided with the drought-prone conditions, i.e. CET-12 > 0, SPI-3 < 0, and SGI-1 < 0 (see the yellow shades). However, not all drought-prone conditions were reported on in the

260 newspapers.

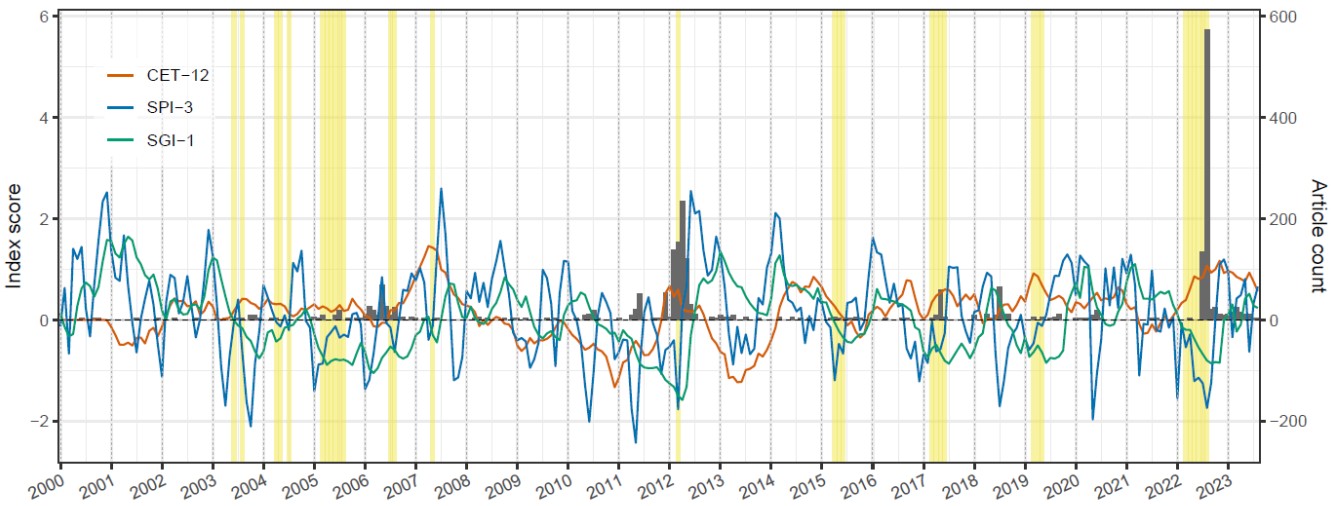

**Figure 1 Time-series plot to show the monthly count of newspaper articles (bar plot) and the hydroclimatic anomalies (i.e., CET-12, SPI-3, SGI-1). Drought-prone conditions, CET-12 > 0, SPI-3 < 0, SGI-1 <0 and spring/summer season, were highlighted in yellow shades.**

265

In Figure 2, we note that both the drought in 2012, which is more strongly related to long-term groundwater deficiency and that of summer 2022, related to a combination of positive temperature anomalies compounded by a precipitation and

groundwater deficit were the subject of most newspaper reporting, and we chose these two events for more in-depth analysis using topic modelling.

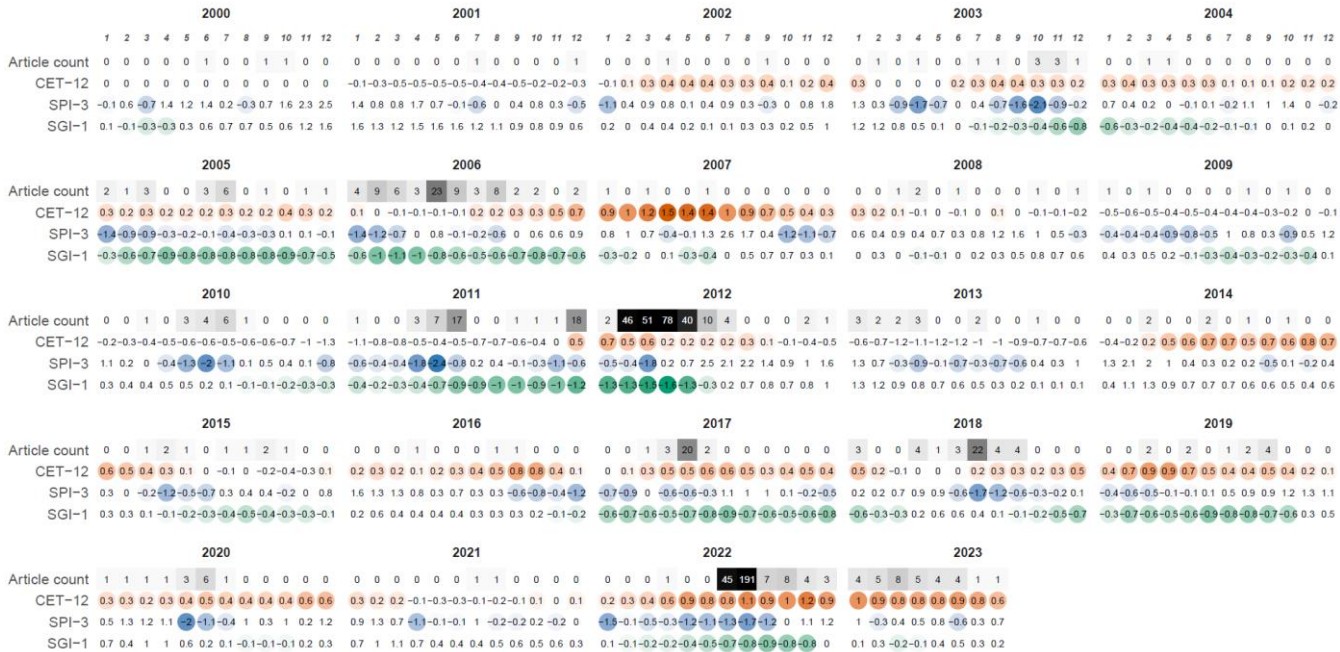

**Figure 2 Time-series bubble plot to show the monthly count of newspaper articles in grayscale grids (dark colours show higher article counts) and the hydroclimatic anomalies (i.e., CET-12, SPI-3, SGI-1) in bubbles (saturated hues indicate drought-prone anomalies).**

## 3.2 Newspaper topics about drought

### 3.2.1 Topic labels in the corpus

According to the coherence and exclusivity test, we confirmed the optimal number of topics, k, to be 15 (see Supplementary material 5). For the 15 topics, we extracted frequent keywords based on probability and frex estimation, together with frequent organization names. As a result, we assigned the topic labels that best describe the keywords and frequent organization names, and summarised the result in Table 1. The order of topics follows the expected probability of topics in the corpus (Supplementary Material 6).

The most common topic in the corpus was "**Drought and hosepipe ban**" (Topic #1), which described hosepipe bans, a popular measure for restricting water use in the UK, due to the severity and expanse of drought status around both meteorological (i.e. rainfall) and hydrological (i.e. river, groundwater) domains. Common organizations in this topic included the Environment Agency, the government body responsible for environment protection and management. Additionally, Thames Water and Southern Water, two large companies to supply water to Greater London and beyond, were identified,

probably because they imposed the hosepipe ban. Interestingly, we found the topic of "**Temperature and hosepipe ban**" (Topic #11) as the 11<sup>th</sup> most popular topic in the corpus, which also described the declaration of hosepipe ban, but in the context of temperature, not rainfall or water reserves.

The second most common topic was "**Water shortage**" (Topic #2), which contained the shortage in both meteorological (i.e. rainfall, rain) and hydrological (i.e. reservoir) droughts, yet not in the phase of announcing water restrictions, e.g. hosepipe bans. Frequent organizations other than the Environment Agency included the Met Office, which is responsible for weather forecasting in the UK. Interestingly, we found the fourth common topic to be "**Flooding**" (Topic #3) seemingly unrelated to droughts. However, this topic addresses how the end of major droughts is reported – often contrast with
periods of sustained rainfall and associated flooding (see Parry et al., 2013). Similar to the second topic that described weather conditions, this topic also featured the Environment Agency and the Met Office as major organizations.

The third common topic was "**Water management**" (Topic #3), which described current managerial concerns regarding water supply companies, as shown in the keywords of 'leak' and 'leakage'. Accordingly, the topic captured frequent organizations to be major water suppliers, such as 'Thames Water', and the 'Ofwat', i.e. Water Service Regulation Authority
in the UK.

The fifth topic was "**Water use impacts** (from hosepipe ban)" (Topic #5). This topic described the range of impacts on daily water consumption when it comes to hosepipe bans, such as restricting water use for 'watering the garden' or 'car wash'. As hosepipe bans are announced by water suppliers, common organizations in this topic included 'Thames Water', 'South East Water', and 'Southern Water'.  Less commonly, but we found topics that describe the broader impacts of droughts,
as shown in "**Agriculture**" (Topic #7), which addressed the impact of droughts on 'crop', 'vegetable', and 'price', by delivering the voices of farmers (i.e. National Farmers Union) as key organization. Furthermore, we found the topics of "**Aqua ecosystems**" (Topic #8) and "**Terrestrial ecosystems**" (Topic #15), which described the drought impact on 'wildlife', 'wetland', and 'tree', with their vocal organization to be the Environment Agency.

The sixth popular topic was "**Heatwave**" (Topic #6), which described 'high temperature', resulting in 'heatwave' and
'(wild)fire'. The most frequent organization in this topic was again the Met Office. We also found discussions related to "**Climate change**" (Topic #10) based around keywords such as 'scientists' to global 'warming' or 'changing climate pattern(s)'. In addition to the Met Office and the Environment Agency, we found the European Environment Agency (EEA) in this topic. The topic of "**Beavers**" (Topic #14) was rare, but interesting given current narratives related to rewilding of this species in the UK (Holmes et al., 2024). We assume these articles link beavers and their activities to drought alleviation,
through for example dam building and wetland restoration.

As is common in topic modelling some topics were not assigned labels (Topics of #9, #12, #13) since they were hard to interpret or irrelevant (e.g., Topic #9 seemed to include 'drought' in sport, especially football, and is therefore irrelevant). Topic compositions for the entire study period can be found in Supplementary Material 7.


**Table 2 Summary of 15 topics with their topic labels, probability keywords, frex keywords, and key organization names. The order of topics follows the expected topic proportions in the corpus.**

| # | Topic labels | Probability-based keyword (prob), frex keywords (frex), and frequent organization names (org). *In descending order of frequency* |
|---|---|---|
| 1 | Drought and hosepipe ban | • (prob) drought, water, ban, hosepipe, company, level, area, rainfall, river, part<br>• (frex) restriction, status, groundwater, ban, hosepipe, drought, environment, agency, area, level<br>• (org) the Environment Agency, Thames Water, Southern Water |
| 2 | Water shortage | • (prob) year, cent, per, water, rainfall, reservoir, month, level, rain, drought<br>• (frex) per, cent, average, reservoir, mm, capacity, rainfall, figure, month, fear<br>• (org) the Environment Agency, the Met Office, Thames Water |
| 3 | Water management | • (prob) water, company, supply, year, leak, drought, reservoir, leakage, plan, government<br>• (frex) leak, target, leakage, meter, bill, metering, regulator, plan, industry, consumption<br>• (org) Ofwat, Thames Water, the Environment Agency |
| 4 | Flooding | • (prob) rain, flood, flooding, drought, weather, week, water, warning, downpour, area<br>• (frex) flooding, flood, downpour, flash, wind, subsidence, tornado, claim, road, sandbag<br>• (org) the Environment Agency, the Met Office |
| 5 | Water use impacts from hosepipe ban | • (prob) water, ban, hosepipe, plant, garden, order, drought, car, people, company<br>• (frex) drip, order, butt, sprinkler, lawn, watering, hose, can, car, swimming<br>• (org) Thames Water, South East Water, Southern Water, the Environmental Agency |
| 6 | Heatwave | • (prob) temperature, day, week, heatwave, weather, water, fire, heat, condition, area<br>• (frex) thunderstorm, temperature, barbecue, heatwave, fire, firefighter, blaze, wildfire, crew, high<br>• (org) the Met Office, Thames Water, Southern Water |
| 7 | Agriculture | • (prob) farmer, crop, drought, food, price, year, weather, potato, condition, vegetable<br>• (frex) potato, wheat, harvest, livestock, barley, grower, crop, yield, vegetable, carrot<br>• (org) NFU (National Farmers Union), the Environmental Agency, EU |
| 8 | Aquatic ecosystems | • (prob) water, river, drought, wildlife, year, fish, level, wetland, environment, bird<br>• (frex) trout, vole, wetland, fish, salmon, stream, wildlife, lapwing, chalk, oxygen<br>• (org) the Environment Agency, Thames Water, RSPB, WWF |
| 9 | n/a | • (prob) garden, year, plant, drought, flower, gardener, thing, goal, grass, day<br>• (frex) prairie, goal, dahlia, trophy, game, show, rose, defender, euphorbia, verbascum<br>• (org) Chelsea, RHS, Arsenal, Tottenham |

| 10 | Climate change | • (prob) climate, change, year, weather, drought, scientist, pattern, summer, world, warming<br>• (frex) scientist, climate, pattern, change, warming, study, emission, extreme, strawberry, world<br>• (org) the Met Office, the Environment Agency, EEA (European Environment Agency) |
|---|---|---|
| 11 | Temperature and hosepipe ban | • (prob) water, ban, customer, hosepipe, company, drought, temperature, area, day, supply<br>• (frex) customer, leakage, bottle, property, bin, litre, health, wildfire, fire, amber<br>• (org) Thames Water, the Environment Agency, Government |
| 12 | n/a | • (prob) water, summer, week, drought, day, rain, weather, people, temperature, heat<br>• (frex) ladybird, standpipe, street, championship, memory, bath, swarm, trial, clock, umpire<br>• (org) Thames Water, the Environment Agency |
| 13 | n/a | • (prob) year, cent, per, price, market, time, school, sale, month, beach<br>• (frex) jellyfish, fever, market, share, pollen, sufferer, investor, school, analyst, beach<br>• (org) Straight, the Met Office, the Environment Agency |
| 14 | Beavers | • (prob) year, water, beaver, drought, people, project, infrastructure, population, reservoir, country<br>• (frex) beaver, cave, fund, bog, dam, peat, project, component, developer, debt<br>• (org) Thames Water, Ofwat |
| 15 | Terrestrial ecosystems | • (prob) tree, drought, year, leave, summer, plant, water, autumn, flower, robinia<br>• (frex) robinia, bluebell, tree, pseudoacacia, ivy, leave, leaf, blossom, blackberry, sapling<br>• (org) the Environment Agency, Labour |

### 3.2.2 Close reading for spring 2012 and summer 2022

For the droughts of spring 2012 and summer 2022, we visualized topic compositions using treemaps (Figure 3) to allow explore differences between topic use for these two events. The most distinctive difference between the spring 2012 and summer 2022 was the dominant topics: for the drought in spring 2012, "**Flooding**" (Topic #4) was the most visible, whereas the drought in summer 2022 highlighted the topics of "**Drought and hosepipe ban**" (Topic #1). For the topics in summer 2022, we found two distinctive topics: first, newspaper attention to the drought in summer 2022 was attributed to the summer

season, as highlighted by topics of "**Heatwave**" (Topic #6) and "**Temperature and hosepipe ban**" (Topic #11); second, we observed the emergence of "**Water management**" (Topic #3) as a prominent topic in the summer of 2022.

     For the most dominant "**Flooding**" (Topic #4) in spring 2012, the newspaper articles positioned the flooding in contrast to the precedent droughts, e.g. "From drought to flood warnings in a week [Article 475, April 2012]", "April showers lead to floods but there's no end to drought [Article 77, April 2012]", and "How drought can lead to flooding [Article 559,

May 2012]". This topic also appeared in the drought of summer 2022, "UK weather: warning of floods from thundery showers after drought [Article 867, August 2022]", but not as strong as spring 2012.

"**Drought and hosepipe ban**" (Topic #1) was the most prominent topic in summer 2022, yet it appeared in spring 2012 as well. The topic describes the onset of drought-prone conditions and the (possibility of) declaration of hosepipe ban, e.g., "Crippling drought hits south and east of England [Article 5, February 2012]" and "Drought fears after England suffers

driest spell since '76 [Article 254, July 2022]". Water suppliers often appeared in the headlines related to this topic, e.g. "Southern Water announces hosepipe ban over drought fears [Article 764, July 2022]". An accompanying topic, "**Water use impacts**" (Topic 5) discusses how people should be prepared when it comes to the drought and hosepipe ban, "Survive the drought and keep your garden going until the rain arrives [Article 787, March 2012]", "Can I wash my car during a hosepipe ban? When you CAN use your hose during drought [Article 1042, August 2022]".

Meanwhile, "**Temperature and hosepipe ban**" (Topic #11), which attributed high temperatures as a trigger for drought and hosepipe ban, appeared exclusively for summer 2022, with the headlines of "Drought expected to be declared amid heatwave with UK hotter than Caribbean [Article 251, August 2022]". In a similar context of summer season narratives, "**Heatwave**" (Topic #6) was visible in the summer 2022, implicating high temperatures as the driver of droughts, heatwave, and wildfire, as shown in the headlines, e.g., "Britons take to the beach as the country basks in 35C heat this weekend as

drought is officially declared [Article 1036, August 2022]" and "Now we face triple threat of heatwave, fire and drought [Article 1231, August 2022]".

The topic of "**Water management**" (Topic #3) focussed on responsibility during the drought of summer 2022. Articles often pointed out the failure in managerial bodies to handle droughts, "Ministers accused of having no plan to deal with drought [Article 635, August 2022]", "Why is Britain not better prepared for this drought? [Article 953, August 2022]".

Furthermore, the narrative deepens with discussions around water suppliers for their managerial failure to handle water inefficiency, e.g., "Water firms fail on leak targets as drought looms [Article 908, August 2022]".

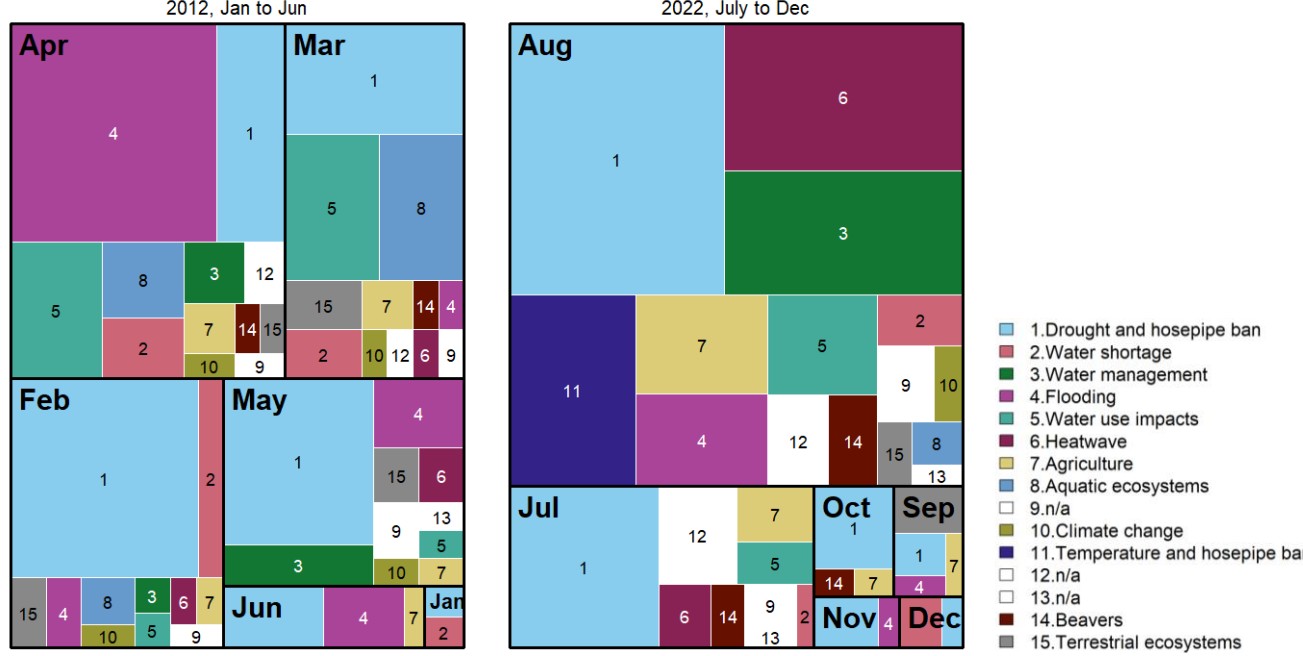

**Figure 3 Treemaps comparing topic compositions for droughts in spring 2012 and summer 2022. The counts of monthly articles are 2, 46, 51, 78, 40, 10 for January to June, 2012, and 45, 191, 7, 8, 4,3 for July to December, 2022.**

## 4 Discussion

### 4.1 How do hydroclimatic conditions relate to newspaper reporting?

Our findings confirmed that newspaper reporting about droughts was related to the conditions when the precipitation (SPI-3) was low (or SPI-12 in the case of the parsimonious model), the groundwater table (SGI-1) was low, and temperature (CET-12) was high, and particularly when the season is spring or summer. Our results are novel in eliciting combinations of hydroclimatic variables and seasonality, which explain the volume of newspaper reports, and also in calculating the scale of contribution of explainable variables on the number of articles, specifically one unit decrease of SPI-3 or SGI-1 resulted in ~50 per cent or ~140 per cent of newspaper article being more published.

While we confirmed that such drought-prone conditions are positively correlated to newspaper attention, our findings also suggest that precipitation anomalies alone are insufficient to explain the volume of newspaper attention and point to the importance of seasonality as an additional factor. This suggests a perception gap with respect to drought between media and scientists, as hydrologists and meteorologists emphasise precipitation and groundwater deficits when defining droughts (e.g., the drought reports from UK CED). Similarly, previous studies examining long-term newspaper reporting of droughts have predominantly focused on precipitation-related indices, such as the SPI (Dayrell et al., 2022; O'Connor et al., 2022). Our best fitting model included temperature anomalies as a factor, and our most parsimonious well-fitting model only included SPI and

seasonality. This is interesting because higher temperatures anomalies are associated with higher potential evapotranspiration, increasing the possibility and the magnitude of hydrological droughts. Indeed, high temperature anomalies tended to garner newspaper reporting about climate change (Pianta & Sisco, 2020). However, such a weak contribution of temperature on newspaper reporting about droughts highlights the precipitation deficit together with low groundwater levels to be more critical factor to perceive the droughts to be newsworthy.

While seasonality is often overlooked in related works, our findings confirmed that given identical hydroclimatic conditions, spring and summer seasons resulted in almost 200 per cent increase (190%, 203%) in newspaper articles, compared to autumn as a baseline. Similar results have been reported by a few researchers, for example O'Conner et al. (2022) noted that seasonality was a meaningful variable in driving the volume of newspaper reporting of droughts, with summer seasons leading to more articles with only a modest SPI-3 deficit. Parsons et al. (2019) suggested that perceived impact of droughts, especially agricultural droughts in warm seasons of July and August, corresponded to more documents in a corpus of farming periodicals in the UK. Such a seasonality bias, although it can be potentially reflecting public interest, newsworthiness, or the scale of actual damages, highlights the risk that early signs of drought during colder seasons may be overlooked by newspapers, thereby failing to generate support for proactive actions or preventive measures.

Two peaks of newspaper reporting about droughts (i.e. spring 2012, summer 2022) overlapped with drought-prone conditions. However, reversely, such drought-prone conditions did not always generate noticeable newspaper reporting as few articles were found in 2003, 2005, 2017, and 2018-2019. Interestingly, these years of drought-prone conditions are confirmed by the UKCEH to publish reports about unusual hydrological events, for the years of 2003 (Marsh, 2004), 2004-2006 (Marsh, 2007), 2010-12 (Marsh et al., 2013), and 2018-19 (Turner et al., 2021). In other words, our analysis of drought-prone conditions corresponded to the scientific acknowledgement of drought conditions by UKCEH, but media did not always pay attention to such occasions. Such a selective nature of media attention was also identified by Dayrell et al. (2022) from their text corpora about droughts over the last 200 years in the UK.

This mismatch, or more precisely the selective nature of newspaper reporting about droughts, may be explained by the newsworthiness of events: even if the meteorological conditions are favourable for droughts, they may not generate media coverage. As explained earlier, droughts are complex, lagged processes with precipitation anomalies linked to deficits in streamflow or groundwater and in the end, the water shortage. Thus, the observation of 'scientific' droughts does not immediately imply direct impacts on people, and thus incipient droughts are rarely the subject of media coverage (Dayrell et al., 2022). For example, the drought of 2004-2006 is observed as an unusual drought event by UKCED (Marsh, 2007), but the actual impact of the drought was minimal (Tanguy et al., 2021).

We discuss two more limitations in our approach that may also contribute to such mismatches: temporal windows and spatial granularity. As temporal windows, we considered 3, 6, 12 months to differentiate the long-term effects of low precipitation (SPI) and high temperature (CET). However, a fixed-window approach may not be suitable to explain all the fluctuations of newspaper reporting, because droughts events, or more specifically newspaper attention to droughts, are uniquely situated: they can be intensified as a result of the duration of drought-prone conditions or it can be extinguished from

the condition of just preceding months, especially the rainfall. For example, 2003 saw severe drought across continental Europe, with the UK also experiencing high temperature and low precipitation anomalies, marked by the second lowest February-October rainfall in 83 years. Nonetheless, English newspaper coverage in this period is not visible, likely because heavy rainfalls in the precedent winter, i.e., November and December of 2002, filled up reservoirs and recharged groundwater resources (Marsh, 2004). Similar patterns can also be found in 2005 and 2019, again leading to limited newspaper reporting with respect to incipient drought conditions. Conversely, the two peaks of newspaper reporting in spring 2012 and summer 2022 were accompanied by a years-long duration of drought-prone conditions, i.e., before spring 2012, there had been persistently low precipitation since mid-2010, leading to a deficit in groundwater, soil moisture, and leaving major reservoirs unfilled (Marsh et al., 2013). Likewise, for the summer of 2022, low precipitation anomalies commenced in the autumn season of 2021 and persisted until the summer of 2022, and hotter temperature anomalies resulted in severe droughts, calling for substantial newspaper reporting.

Another limitation of our study is its spatial granularity. When processing the indicator metrics, we aggregated the variables nationally (i.e., across England), obscuring regional variations or temporal transitions over geography. Although in principle it would be possible to assign news articles to smaller regions using methods from geographic information retrieval (Acheson & Purves, 2020), in practice this would further stratify our corpora, making statistical modelling challenging. To consider geographic variance better, which is important because droughts are geographically heterogeneous (Tanguy et al., 2021), a larger corpus would be necessary (Parsons et al., 2019). To overcome this limitation, it would also be necessary to prepare a complete set of documents that can be assigned to finer scales, e.g. catchments, corresponding to the scale of meteorological stations that generate hydroclimatic metrics (O'Connor et al., 2022).

## 4.2 Which topics are represented in newspapers?

Using topic modelling, we identified 12 meaningful topics in drought-related newspaper corpus and visualised their compositions over time. By carrying out close reading we zoomed into the detail of newspaper narratives for two major drought events in the newspaper: spring 2012 and summer 2022. This multi-scalar method allowed us to take advantage of both methods, i.e., computational topic modelling elicits the overarching composition of topics and close reading complement the in-depth context of these topics in the corpus. As a result, this study enabled to distinguish between the newspaper narratives between two major droughts in England.

From the topical composition, we found the topic "**Drought and hosepipe ban**" (Topic #1) was present in almost all months. The topic encompasses meteorological and hydrological droughts in, e.g. rainfall and river, and further reports on both potential and actual declarations of hosepipe ban, a common management measure to reduce water usage in England, imposed by water companies. This highlights on the one hand, the role of media in informing readers about drought conditions that may pose significant impacts on daily lives, and on the other hand, in emphasizing readers concerning water supply and restrictions when it comes to droughts. According to Stahl et al. (2016), a major concern related to droughts in the UK was public water

supply, which contrasts with other countries where different concerns, e.g., drinking water quality or agriculture and livestock farming, emerged. Stahl et al. (2016) also revealed that 'local and regional water supply shortage and problems (drying up of springs/wells, reservoirs, streams)' and 'bans on domestic and public water use (e.g., car washing, watering the lawn/garden, irrigation of sports fields, filling of swimming pools)' consisted of major water concerns in the UK. These findings are very

much in accord with the results of our topic modelling.

Repeated droughts and hosepipe ban in England seemed to have resulted in more discussion of water management, as shown by the topic "**Water management**" (Topic #3), which became much visible in the drought of summer 2022. Such a pattern can be ascribed to the high volume of media reports in August 2022, scoring 191 articles for a single month. At the same time, close reading alluded that repeated droughts and hosepipe ban, at least in the last two decades, have galvanized

newspaper to redirect their attention to the water management applied by government and water suppliers. The results suggest growing criticism towards water supply companies being unable to handle metering or leakage issues, compounded with the privatization of water supply in England (Bayliss et al., 2021).

Topics related to "**Heatwave**" (Topic #6) and "**Temperature and hosepipe ban**" (Topic #11) seem to point to some confirmation bias concerning newspaper reporting. Both topics were very prominent in the summer of 2022 and through close

reading, we concluded that these topics gravitated to typical summer narratives, e.g., relating high temperatures and heatwaves to drought conditions. Our negative binomial regression modelling also confirmed a seasonality bias – with spring and summer seasons resulting in increased newspaper reporting.

One important limitation of our content analysis lies in the nature of our corpus, which is restricted to a single genre: newspapers. Despite the advantages of newspapers as a data source (Boykoff, 2008), this genre has experienced significant

declines in readership given the rise of online news platforms (Kim et al., 2019) and social media platforms such as Twitter/ X (Antwi et al., 2022). Extending the corpus to incorporate more genres of writing about drought, e.g., by incorporating resources including the UK Drought Inventory (e.g., Parsons et al., 2019), and the European Drought Impact Inventory (EDII) (e.g., Stahl et al., 2016), as well as social media could provide valuable additional perspectives.

## 5 Conclusion

In this study, we constructed a corpus of newspaper articles reporting on droughts in England to analyse: (1) how newspaper reporting correlates with hydroclimatic conditions and seasonality and (2) the topics that emerge in newspaper narratives. Our statistical analysis confirmed that newspaper reporting is strongly influenced by low groundwater levels (SGI-1) and low precipitation levels (SPI-3). Interestingly, temperature (CET-12) had a negligible impact on garnering newspaper reporting, while seasonality, particularly warm seasons such as spring and summer, played a critical role in generating more

media coverage. Using topic modelling we could zoom into newspaper narratives on two severe drought events in spring 2012 and summer 2022. The results showed that newspaper narratives about droughts often gravitated towards drought reports on meteorological and hydrological droughts together with hosepipe ban. On top of that, the summer drought of 2022 was

interesting to reveal summer-related topics, underscoring a seasonality bias in drought reporting. This study contributes to
further our understanding of how newspaper reporting dynamics are intertwined with hydroclimatic conditions and the
perception of seasonality, and how the narratives fluctuated over the last 24 years. We suggest paying more attention to this
potential mismatch between scientific definitions of droughts and their reporting in the media, since the latter play a very
important role in shaping public attitudes and, in turn, the likely effectiveness of policy interventions.

## Code availability

The R code lines we applied for the analysis were archived in a Markdown file, which can be found in the Supplementary
Material 1.

## Supplementary Materials

All the supplementary materials mentioned in the manuscript, including the R Markdown, can be found in the online version
of publication.

## Author contribution

All the authors (IK, JS, RP) conceptualised and developed the research questions. IK took a lead on implementation, including
data collection, formal analysis, and visualisation, upon the communication with all authors. JS and RP prepared the original
draft, and all the authors (IK, JS, RP) contributed the edits, reviews, revisions for the final draft.

## Competing interests

One of our co-authors is a member of the editorial board of Hydrology and Earth System Sciences (HESS).

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
