# Peer review of "Droughts and Media: when and what do the newspapers talk about the droughts in England?"

_EGUsphere, 2024_

## Author Response (AR1)

**Response to three reviewers' comments**

**Droughts and Media: when and what do the newspapers talk about the droughts in England?**

**Notification on major change**

We would like to thank all three reviewers and the editor for taking the time to read and provide thoughtful comments on our manuscript.

Before addressing the detailed comments line by line, we would like to outline the major changes made in our revised manuscript. These changes address shared concerns raised by more than one reviewers and we assured that these will enhance the overall integrity and robustness of our work.

In the first part of our analysis, we aimed to examine the relationship between explainable variables and the volume of news media attention. Originally, this analysis involved precipitation and temperature anomalies, assessed alongside a chi-square test to explore seasonal variations. In the revised manuscript, we replaced the precipitation anomaly with WMO drought indices (SPI/SSI) and incorporated additional variables such as SGI and seasonality. Consequently, we transitioned from using chi-square analysis to negative binomial regression analysis. This methodological change allowed us not only to identify the best-fitting combinations of variables in explaining the volume of newspaper articles but also to quantify the contribution of each variable.

The second part of our analysis applied a multi-scalar text analysis approach, combining computational topic modeling with qualitative close reading. While the methodology in this section largely remains unchanged, we increased the number of topics from 10 to 15, based on coherence and exclusivity tests. This adjustment enabled us to uncover rarer but meaningful topics within the corpus, which further enriched our findings. For example, this revision illuminated the role of seasonality in influencing media attention, bridging insights from the statistical analysis.

These changes would make some comments from the reviewers no longer applicable. Below, we present detailed responses to the comments from the three reviewers. Thank you for your time and consideration.

**RC1.**

**Notification on major change**

We would like to thank all three reviewers and the editor for taking the time to read and provide thoughtful comments on our manuscript.

Before addressing the detailed comments line by line, we would like to outline the major changes made in our revised manuscript. These changes address shared concerns raised by more than one reviewers and we assured that these will enhance the overall integrity and robustness of our work.

In the first part of our analysis, we aimed to examine the relationship between explainable variables and the volume of news media attention. Originally, this analysis involved precipitation and temperature anomalies, assessed alongside a chi-square test to explore seasonal variations. In the revised manuscript, we replaced the precipitation anomaly with WMO drought indices (SPI/SSI) and incorporated additional variables such as SGI and seasonality. Consequently, we transitioned from using chi-square analysis to negative binomial regression analysis. This methodological change allowed us not only to identify the best-fitting combinations of variables in explaining the volume of newspaper articles but also to quantify the contribution of each variable.

The second part of our analysis applied a multi-scalar text analysis approach, combining computational topic modeling with qualitative close reading. While the methodology in this section largely remains unchanged, we increased the number of topics from 10 to 15, based on coherence and exclusivity tests. This adjustment enabled us to uncover rarer but meaningful topics within the corpus, which further enriched our findings. For example, this revision illuminated the role of seasonality in influencing media attention, bridging insights from the statistical analysis.

These changes would make some comments from the reviewers no longer applicable. Below, we present detailed responses to the comments from the three reviewers. Thank you for your time and consideration.

**Overall comments**

This paper provides an assessment of drought reporting by collating newspaper articles related to droughts in England check their alignment with meteorological anomalies. The study suggests that media coverage tends to coincides with meteorological drought, but that drought only coincides with media coverage when additional conditions are "favorable" (e.g., seasonality, long-term precipitation shortage). Additionally, it concludes that media often reports about water shortages, water use restrictions, and the consequences that follow.

The first part of the analyses uses statistical tests (chi-squared test) to quantify that most report on droughts occur when precipitation has been below-average and temperatures have been higher than expected (Fig 1) and that these effects tend to be more reported in Spring and Summer compared to other

seasons. These findings are fine but given how trivial they are it also seems extremely marginal for a contribution to HESS (in my opinion).

We appreciate your comment on this matter. In our revision, we adjusted our methodological approach in this matter of examining temporal alignment between the media attention and drought-prone conditions (please see *Major change* at the beginning). Originally, this analysis involved precipitation and temperature anomalies, assessed alongside a chi-square test to explore seasonal variations. In the revised manuscript, we replaced the precipitation anomaly with WMO drought indices (SPI/SSI) and incorporated additional variables such as SGI and seasonality. Consequently, we transitioned from using chi-square analysis to negative binomial regression analysis.

Although the findings may seem 'trivial', i.e., drought-prone conditions resulted in media attention, our result is meaningful to reveal the best-fitting combination of variables to steer the media attention about droughts (SPI-3, SGI-1, CET-12, and seasonality to be included), but also to quantify the contribution of each variable (Table 1).

When these findings are combined with the following analysis with media narratives about droughts, we could harness the argument about the seasonality bias in both the scale of media attention and the narratives within. Overall, our study pointed out the potential mismatch of droughts conceptualized between scientists and media, and the lack of media attention for 'less newsworthy' seasons may hinder the implementation of proactive/preventive measures (Discussion).

Before I can recommend this paper for publication in HESS (or any other significant hydrological outlet) I expect some more depth to this finding. I expect the following points to be addressed thoroughly:

- The paper shows the statistically significant more frequent reporting on drought when there are anomalously dry and hot conditions, and states this can be because "a potential confirmation bias". Can the paper also explore the more logical explanation that people report on droughts when there are droughts? Also can anything be said more than there is a statistically significant relationship?

  In our revised manuscript, we confirmed the seasonality bias in media attention during warmer seasons through both statistical and narrative analyses. In the statistical analysis, the best-fitting model included SPI-3, SGI-1, CET-12, and seasonality as explanatory variables. Using autumn as the baseline, we observed a 189% increase in newspaper articles during spring and a 203% increase during summer. From the topic modeling, we identified two dominant topics in the summer of 2022 related to the season: "Heatwave" and "Temperature and Hosepipe Ban." These findings support a discussion on the potential confirmation bias in media coverage, which may favor generating drought-related articles during warmer seasons.

- Can it be more robustly reported how likely it is that (according the scientific definition) drought events are actually picked up significantly by the media, and how this differs between different

types of drought, different types of drought intensities, and differs per season. Hereby can the statistics that express this go beyond a chi-squared test that only tests whether two variables are independent or not. The plots 3 and 4 try to give some more depth but are only qualitative suggestions and no formal tests or numbers that quantify this.

In our revised manuscript, the new statistical analysis enabled us to identify the best-fitting combinations of variables to explain the volume of media attention. Using this combination, we estimated the contribution of each variable to the observed media attention.

However, we were unable to differentiate between different types or intensities of droughts, largely due to the limited number of articles beyond the two major drought events. Additionally, we applied static temporal windows (e.g., SPI-3, SPI-6, SPI-12), representing fixed moving windows of 3 months, 6 months, and 12 months, which have limitations in capturing the dynamics of preceding conditions (on this matter, further elaboration can be found in the Discussion 4.1, 6th paragraph). Nonetheless, our topic modeling allowed us to compare the topical composition between spring 2012 and summer 2022, revealing a stark contrast in media narratives (refer to Chapter 3.2.2 and Figure 3).

- Optional: can something be said if drought reporting has become more or less frequent over time (i.e. if the same drought event now gets different coverage than in the past?).
  We believe this is a relevant and intriguing question when dealing with large-scale corpus analysis. Unfortunately, our corpus-building time frame was relatively short (24 years). To our knowledge, several studies have analyzed documents over longer periods, such as Dayrell et al. (2022), which examined a 200-year news text corpus on drought and rainfall in the UK, and O'Connor et al. (2023), which analyzed a corpus spanning over 100 years in Ireland. These studies had to account for the increase in publication volume in recent years, driven by advances in publishing and text archiving.

- How do we know that news items actually are about a current drought and do not reflect on conditions in the longer past? How does this affect the analyses? Can this be screened for?
  This is a good question. In practice, computational applications of large-scale data often rely on an assumption—for example, that people are discussing current events. While this assumption is generally applicable to 'news' articles, given their purpose of disseminating the latest updates, we recognize that it may not always hold true, such as with newspaper articles revisiting past events. During the revision process, we had the opportunity to manually review a subset of articles through close reading, and we found such cases to be almost nonexistent. Instead, our filtering process revealed that handling irrelevant articles posed a greater challenge, as reflected in the remaining n/a topics in Table 2, even after the data cleaning process.

- The paper interprets that more common reporting of summer droughts "could indicate a confirmation bias driving media coverage –drought is more likely to be reported in warmer conditions since it confirms a simplistic narrative relating drought to warm weather." Is an

alternative considered where maybe there is just a mismatch between the scientific definition of the drought used, and the drought society and reporters care about (when they actually feel impacts will be during summer drought and not really during "droughts" in for example winter, because then, while conditions are anomalous the impacts on society will be absent or minimal.. You already acknowledge this yourself in the paper and therefore also omit data from other regions of the UK.

We agree with your observation and acknowledge this important aspect. We do not claim that confirmation bias is the sole explanation, but rather one possible factor. In the revised manuscript, our new statistical analysis allowed us to test various combinations of explanatory variables, confirming that seasonality is indeed correlated with the volume of media attention. Our results reaffirm that precipitation deficits contribute to driving media attention (aligning with the 'scientific' conceptualization of drought), but we also found that warm seasons exert a statistically significant influence on generating increased media coverage. Additionally, the topic modeling of the summer drought of 2022 further supported the seasonality bias, as evidenced by the prominence of summer-oriented topics.

- The paper is about drought and reporting on drought, but ends up using an index based on accumulated weather conditions (P and T) and not a regular drought index. This seems to add an additional layer of inconsistency in the analysis as right now it balances between droughts people likely care about (bc they are reported in the news), drought as hydrologists typically define (e.g, standardized indices), and droughts as defined in this paper (P and T anomalies). This additional step op not using a typical drought index makes it harder to connect the findings to the existing literature on drought

Thank you for your comment. Indeed, we received similar comments from other reviewers, and we took this as one of major changes (please refer to Major changes at the beginning).

- The part were drought are assessed by their specific keywords. Can the authors express what we truly hydrologically learn here and how this is a scientific contribution that fits within the scope of HESS? ("HESS encourages and supports fundamental and applied research that advances the understanding of hydrological systems, their role in providing water for ecosystems and society, and the role of the water cycle in the functioning of the Earth system.") I personally struggle to see how the contributions of this section align with these objectives of HESS.

We submitted this paper to a special issue in HESS titled **Drought, society, and ecosystems**. In the call, the editors state that "*we aim to showcase the diverse interdisciplinary research being done on the interactions between drought, ecosystems, and people (including human-induced climate change and management)*". Furthermore, the topics listed include drought risk management and communication. We believe that our work is interdisciplinary and directly addresses the topic of communication by and to non-experts, which is in turn very important in understanding and developing future policy. Mismatches between scientific discourses and popular understandings are central to challenges in changing behaviour more broadly. We also think that work like ours

fits to HESS more broadly, in terms of the link to water and society, but we explicitly prepared and submitted the paper to this very relevant special issue.

- The statistical tests (chi-squared test) assumes all observations are independent. To what extent is this the case in the dataset of this paper?
  In our revised manuscript, we replaced the statistical analysis from chi-squared test to negative binomial regression analysis. Moreover, prior to running the model, we ran a pair-wise correlation analysis to test the multi-collinearity. As a result, we could not include SSI, since it correlates strongly to SPI.

**Detailed comments**

L12-14: this final sentence of the abstract is unclear in its meaning (to me). Please check.

We reworded the abstract to clarify our findings and revised it to reflect the changes in methods and results.

L21: Tjideman is misspelled and not listed in the reference list…

Thank you for finding this. We corrected this in the manuscript.

L25: I am unsure this the universal case for what a drought is. In some cases the balance of a drought will also be influenced by e.g. how much discharge has drained the system. In addition, the sentence after already starts to contradict the statement in line 25.

We reformulated this paragraph (2nd paragraph in the Introduction) to state the definition of droughts, different types of droughts, and droughts to be a complex phenomenon to comprehend. In the next paragraph, we continue on addressing the research gap and our research questions.

RQ1. How does the media coverage relate to meteorological conditions and seasonality in England?" "How […] relates" is pretty vague and not reflects any rejectable hypothesis. Maybe consider phrasing this bit more precise.

We rephrased this research question as follows:

*RQ1. How and to what extent do hydroclimatic conditions and seasonality relate to newspaper reporting in England?*

As such, we meant to measure a) how, i.e., whether the variables are positively or negatively related to the volume of media attention, and b) to what extent, i.e. how much the variables can explain the volume of media attention. Such a change was possible since we introduced regression analysis in the revised version.

RQ2. What are typical topics in media content, and how do they differ for different major drought events? This question needs be rephrase for clarity (what media content are you talking about, can "typical topics" be defined more clearly" etc).

Likewise, we rephrased this research question as follows:

*RQ2. What are the dominant narratives in newspaper reporting of droughts, and how do these narratives vary across major drought events?*

We believe new phrase of research question serves our analysis of topic modeling and close reading, which aimed to grasp dominant media narratives and then dive into the major drought events of spring 2012 and summer 2022.

Indeed, these two research questions are not designed for testing hypotheses—acceptance or rejection—but rather to reflect the conceptual foundation and motivation behind this analysis.

**RC2.**

**Notification on major change**

We would like to thank all three reviewers and the editor for taking the time to read and provide thoughtful comments on our manuscript.

Before addressing the detailed comments line by line, we would like to outline the major changes made in our revised manuscript. These changes address shared concerns raised by more than one reviewers and we assured that these will enhance the overall integrity and robustness of our work.

In the first part of our analysis, we aimed to examine the relationship between explainable variables and the volume of news media attention. Originally, this analysis involved precipitation and temperature anomalies, assessed alongside a chi-square test to explore seasonal variations. In the revised manuscript, we replaced the precipitation anomaly with WMO drought indices (SPI/SSI) and incorporated additional variables such as SGI and seasonality. Consequently, we transitioned from using chi-square analysis to negative binomial regression analysis. This methodological change allowed us not only to identify the best-fitting combinations of variables in explaining the volume of newspaper articles but also to quantify the contribution of each variable.

The second part of our analysis applied a multi-scalar text analysis approach, combining computational topic modeling with qualitative close reading. While the methodology in this section largely remains unchanged, we increased the number of topics from 10 to 15, based on coherence and exclusivity tests. This adjustment enabled us to uncover rarer but meaningful topics within the corpus, which further enriched our findings. For example, this revision illuminated the role of seasonality in influencing media attention, bridging insights from the statistical analysis.

These changes would make some comments from the reviewers no longer applicable. Below, we present detailed responses to the comments from the three reviewers. Thank you for your time and consideration.

**Overall comments**

The authors of this study developed a database of drought-related newspaper and tabloid articles in England from 2000 to 2023. Using this database the authors then looked at how the number of articles in the database varied seasonally. They found that the largest number of articles occurred in typically warmer and dryer months. Next, the authors used an unsupervised natural language processing method called topic modelling to cluster articles in the databases into 10 topic areas. The clusters that emerged covered topics that all intuitively related to drought. While the results are not all that surprising, I think the methods and the presentation (i.e., figures) are interesting and would contribute a unique perspective to HESS.

Thank you for the comment. In our revised manuscript, we extended our analysis, both statistical analysis and topic modeling analysis, to bring further and richer contribution to HESS. Please refer to ***Major change*** at the beginning of the response.

First, I recommend the introduction be restructured to improve flow. For example, the paragraph starting on line 44 starts with a topic sentence summarizing previous methods used to explore media coverage responses to extreme weather events; however, this paragraph ends with a discussion of the role of newspapers. I think this newspaper discussion could be moved to another part of the introduction (maybe earlier?) and the authors could more concisely review all the methods that have been used to do this type of work. Additionally, by the end of Line 69, I'm left wondering what are the missing (methodological) pieces that this study aims to fill? I think the authors need to be more direct here. What are they doing differently compared to the other studies they list? Last, I'm not seeing any description of topic modelling in the introduction, which seems like an important component of the study that the authors point out in the abstract, but don't mention again until the methods section. See my line comments for more suggestions.

We reorganized and restructured the introduction in a way to:

- Address overall research gap (i.e. lack of understanding in the interplay between media and drought) in the 3rd paragraph
- Explain why newspaper is a good material for proxying the society in the 4th paragraph
- Summarise previous works related to the first research question (i.e. temporal alignment analysis) in the 5th paragraph
- Summarise previous works related to the second research question (i.e. news content and narratives analysis) in the 6th paragraph

Second, I recognize that no drought index is necessarily perfect at capturing all aspects of drought, but I recommend the authors including a WMO standardized drought index such as SPI for precipitation as well as EWP in the Supplement. I think this would allow them to compare this study to other studies conducted nearby (e.g., O'Connor et al. 2022) as well as globally. It might also be of interest because of the experimental design of this study; SPI leverages precipitation anomalies. At a minimum, I recommend the authors include a discussion (potentially in the Supplement) of how EWP compares to SPI and how CET compares to other indices calculated across the UK in terms of their behavior, benefits, and challenges. I had some other questions about the methods, which are included in my line comments.

Thank you for the comment. As we received a similar comment about this matter, and elaborated on this at the beginning of this response – please refer to "***Major change***".

Third, the figures were very creative, but some additional information is needed to help the reader understand them. For example, a scale color key is omitted from several images in the main and Supplemental text. Some of the abbreviated axes labels need to be explained further. See my line comments for more details.

Thank you for the comment. We believe the comment about text is referring to Figure 3. We replaced the texts with just numbers in the figure, and added a legend where readers can find the corresponding labels to the numbers.

Last, I wanted to say that I greatly appreciate the authors transparent and reproducible approach; specifically, their willingness to share their R code and R session information. I think that's awesome and was glad to see it.

Thanks a lot - we appreciate your feedback. According to our revision, we updated the code lines and attached to Supplementary Material 1.

Given these general comments as well as my line comments below, this paper needs a great deal of revision before it will be ready for publication. Therefore, I recommend this manuscript be accepted with major revisions and resubmitted to HESS.

Thanks to the authors for this interesting article and for the opportunity to review this manuscript.

Thank you so much for your feedback.

**Detailed comments**

Line 6. Write out UK, as in "The United Kingdom (UK)…". I know this is a pretty common abbreviation, but to make all readers are on the same page.

We clarified this abbreviation in the first line of introduction. In the abstract, we did not use UK at all, so the abbreviation was not explained here.

Lines 7-9. I recommend the authors include the duration and the number of articles included in the corpus somewhere around here.

Thank you for pointing this out. This sentence in the abstract now is phrased as follows:

*We constructed a corpus of more than 800 newspaper articles related to droughts in England for the last 24 years (2000 - 2023), and…*

Line 11. In "…but the inverse case is not always the case…" consider using another word other than "case"; the second occurrence is repetitive. I suggest "but the inverse case is not always true" or something similar.

This statement still holds true in our revised manuscript, but we decided to not to make a comment about it in the abstract. Rather, we emphasize the contribution of hydroclimatic condition and seasonality on the volume of media attention, and how such a finding is confirmed again in topic modeling. Thank you for your comment.

Line 13. Consider adding more descriptors to "dominant topics" so this is clearer. I suggest something such as "Dominant topics that emerged from the analysis of the newspaper article corpus include…".

Thank you for the comment. In our revision, we only mentioned "Drought and hosepipe ban" as the most prominent topic, and briefly mentioned the 'summer-related topics' ("Heatwave", "Temperature and hosepipe ban"), which align to the result of statistical analysis.

Line 14. I didn't know what a hosepipe is and had to look that up. Maybe there's another way to word this so it's more general? I suggest rewording to something like "household water use restrictions" or similar.

Thank you for pointing this out. We restructured this sentence in a way to explain the topic, for readers who may be not familiar with it.

*The following topic modelling allowed us to reveal dominant narratives to be the "Drought and hosepipe ban", which concerns of meteorological and hydrological water deficiency and a common measure to restrict water usage in England. (LL11-13).*

Line 19. UKCEH citation has no date. I think this might be the Turner et al. 2021 citation, right? Please correct this in-text reference.

This citation was meant to refer to the webpage, which did not have a latest update date. Although it is a proper way of making a citation, we agree that it is unusual. In the revised manuscript, we tagged all the individual reports from this website, instead (second sentence of the 1st paragraph in the Introduction).

Line 23. The authors give citations, but please write out how climate change will heighten drought-prone meteorological conditions in England and the UK, in general, over the next 25-50 years. This would also be a great place to cite IPCC AR6 (https://www.ipcc.ch/report/ar6/syr/) or a national climate assessment.

Thank you for your recommendation. We revisited our citations (Dobson et al., 2020; Guerreiro et al., 2018; Kay et al., 2023; Spinoni et al., 2018) as well as the IPCC AR6 report (IPCC, 2023) as you suggested. While these sources were valuable, we could not find specific numbers or phrases that directly addressed the UK or England. Instead, we confirmed a broader trend for "Western and Central Europe (WCE)" and incorporated this into the manuscript. Regarding specific figures, such as the extent to which climate change will exacerbate these conditions, we encountered a range of scenarios, making it challenging to pinpoint exact numbers. Ultimately, we opted to cite the relevant literature, including the IPCC report, to provide contextual background on the climatic trends. The change is made at the last sentence of the first paragraph of Introduction.

Lines 37-38. The statement "only in a few cases lead to" is vague. Can the authors provide more specifics here? Are the few cases just in the UK and if so when? Providing an example such as "(e.g., )" would be helpful here.

In this paragraph, we meant to explain the complexity in the development of droughts, and only a subset of meteorological and hydrological droughts leads to the 'drought' that can be tangible to the public and society. To clarify this point, we rephrased the sentence as follows (you can find it at the end of 2nd paragraph of Introduction):

*Deficits in rainfall (i.e., meteorological drought) do not always coincide with or lead to low streamflow or low groundwater (i.e., hydrological drought), and not all such cases result in tangible impacts for society (i.e. socioeconomic droughts).*

Lines 40-43. "More critically, media coverage is not solely driven by events. Newspapers choose topics likely to interest or impact their readers directly, and other external factors may also influence the reporting of extreme weather, confirming the role of media in agenda setting and framing." What external factors? Please list these/give examples. Is there a reference the authors can cite here to back up this claim about what drives media coverage?

We clarified this point by rephrasing the sentences as follows (you can find this at the beginning of 3rd paragraph):

*Given the complex and gradual nature of drought dynamics, it is crucial to analyse how and when droughts are reported in the media, as these factors play a significant role in shaping public perception and response. The timing of media coverage reflects not just the occurrence of droughts but also the point at which their impacts become newsworthy (Caple, 2018). This raises important questions about whether media reporting aligns with the actual progression of drought events.*

Line 49. Please write out the SPI and SSI acronyms at least once in the text.

We clarified this point when the term appears for the first time in the manuscript, i.e., SPI (L65), SSI (L151), and SGI (L151). Thank you for point it out.

Lines 53-54. "Again, newspapers are typically at the centre of analytic materials, having a large circulation size reaching different population strata (Boykoff, 2008)." Reconsider using "Again" here. I'm not sure what previous statement the authors are referring to as well as what they mean by "analytic materials". Or was that last part supposed to be "Centre of Analytic Materials" (as a proper noun)? That's the first time they mention that center. Overall, this sentence is confusing to me and I suggest the authors rework it.

This comment is reflected in the revised manuscript as rephrasing the sentences/paragraphs as follows (you can find it at the beginning of 4th paragraph of Introduction):

*Despite the rise of new and diverse media forms, most prominently as social media, newspapers remain a vital data source for media analysis. Written by professional journalists and reporters, newspapers often have large circulations reaching broad and diverse segments of the population (Boykoff, 2008).*

Then in the same paragraph, we continue to describe why news media, specifically newspapers, can serve as a good analytic material.

Lines 70-75. Why is there a need to understand public perception of drought? The authors state UK policies but for those not familiar with them, more specifics would be helpful here with examples for the UK (including citations). What does "understanding public perceptions" look like to these authors? Maybe use a more specific word than "understanding"?

This comment is reflected in the new phrasing of 3rd paragraph, which is edited to describe the motivation of this study. Specifically, you can refer to this point:

*Additionally, the framing and narratives constructed in media coverage deserve close examination, as they influence public perception and societal response, from individual behavioural changes (Antwi et al., 2022; Quesnel & Ajami, 2017) to policy initiatives (Hart et al., 2015). Thus, understanding the temporal and narrative dimensions of drought reporting is essential for enhancing public engagement, fostering informed decision-making, and driving effective societal action.*

Lines 78-85. Can the authors briefly say more about the methods they're using here. In the abstract they mention "topic modelling" but I don't see any mention of this in the introduction.

In the Introduction, we tried to make a brief summary of related works, and improved the 6th paragraph of Introduction, to mention of natural language processing, with an example of Bohr (2020) to apply topic modeling.

*However, most studies heavily rely on a qualitative approach, which poses significant constraints in the scalable application of content analysis. In recent years, natural language processing (NLP) has been implemented to process a large set of text data and elicit key narratives, for example, Bohr (2020) to apply unsupervised text analysis with topic modelling about newspapers discussing climate change in the United States, and Sodoge et al. (2023) to apply pre-trained text classifiers to classify drought impacts from German newspapers (2000-2021).*

Also noting that until now, the authors have been talking about the UK but now they focus just on England. I think it's worth specifically pointing out this narrowing of the study area/study focus.

At the end of first paragraph of Introduction, we briefly touched that England is particularly more susceptible to droughts in the entire UK.

*England is more susceptible to severe droughts within the UK due to less precipitation and greater water demand from human influences (Tanguy et al., 2021; Tijdeman et al., 2018).*

My last comment here is with respect to how they use newspaper and media interchangeably. In my mind, media is more general and includes other non-newspaper communication forms (e.g., TV news, magazines, online videos, etc.). Why don't the authors use newspapers since this is more specific and more applicable to their study?

Regarding our work, we replaced the terms 'media' to 'newspaper (articles)' to clarify our key material for the analysis. When referring to literature works, which entail a variety of media sources, e.g., media scripts, social media, and so on, we kept the terminology 'media'. Also, in the discussion, when we expand the implications beyond our scope of study, we used the term 'media'.

Lines 86-94. I suggest moving this text into the end of the introduction.

We put these sentences at the beginning of Method, in a way to make a quick reminder of research questions, and give an overview of our methodological approaches.

Lines 94-95. Can the authors also provide the R version and cite R (i.e., R Core Team see https://ropensci.org/blog/2021/11/16/how-to-cite-r-and-r-packages/).

We added the citation and version information for/of R package at the end of Chapter 2.1.1.

Lines 112 – 117. Can the authors share the time frame of the 836 articles here? It might also be helpful to note how many years and the number of official reports from UKCEH this range included.

Time frame of corpus building is mentioned in original manuscript, L101. In the revised manuscript, you can navigate to the same sentence in LL121-122.

Line 120 – 123. I'm a little confused by the wording here. Did the authors use precipitation and temperature data in addition to the CET and EWP? If they used additional data, please include more on the datasets (e.g., where they came from, how they were used, etc.).

EWP is no longer included in our revised manuscript. Thank you for your comment.

Line 123. What is the authors' definition of "long-term" here? Please include the number of years considered in the development of these indices as some folks might not be familiar with them before reading this paper. How does the behavior of these indices compare to WMO standard indices like the SPI and does the selection of these indices (over other one(s)) impact the results of this study? I was also curious if the CET is on a monthly time step like the EWP? If not, could that impact the results (or not)? Please discuss this.

The term 'long-term' here is to describe the process of calculating temperature anomalies (i.e. CET). We calculated the 'long-term' baseline temperature by averaging CET for 1991-2020 (30 years). Then, this baseline is used to estimate the 'anomalies' of temperature.

EWP is no longer included in our revised manuscript. Thank you for your comment.

Line 128. I'm a little confused here. The authors previously said these were long-term drought indices, but this statement seems to imply they are short-term, "instant". Can the authors please clarify this in the text?

The term 'long-term' was to indicate the baseline for calculating anomalies. Instant here refers to temporal windows of indices, e.g., 3-month window is shorter than 12-month window.

Lines 146-147 and 153-154. What specific cutoff (alpha level/significance level) for this test did the authors use? Please state this.

This analysis is no longer included in our revised manuscript. Thank you for your comment.

Line 160. Can the authors add a citation for this method?

Winsorisation is an old common method for data normalisation, which is also equipped in DescTools in R (Signorell, 2023). We tried to find the reference, but relevant studies we found rarely mention this with a citation. Also, when we apply this, we did not use the DescTools package, which we found it not necessary to cite this R package.

Line 174. Capitalize R here, as in "R package".

We capitalized the letter (L193 now). Thanks!

Line 175. Did the authors use any metadata variables here? If so, what did they use? Please list these in the text and/or provide a table.

We tested several metadata, (using prevalence function in stm package), but we ended up not including this function. For clarification, we deleted this description (L203).

Line 184. I'm not sure what the authors mean by "crisp". Maybe this is a typo?

We meant to say 'a single topic' out of our mixed-membership model. To clarify this, we replaced this term with 'single' (L212).

*Assigning one single topic to each article is helpful in estimating the count and ratio of topics over time.*

Line 186. Authors switch to passive voice here ("was performed"). Please be consistent throughout.

This sentence now looks like this (LL213-214):

*We further extracted entity names and sorted them by frequently mentioned organization names within the sub-corpora.*

Line 194. There's not much discussion of the procedure used to select k = 10 (from the Supplement and in Section 3.2.1) here. I think this is important and should be include in the methods since the topic assignment results hinge on the value of k.

We described this point in the following text (LL204-206):

*Since topic modelling is an unsupervised method, it is necessary to experiment to find the best-fitting number of topics, k, and we used two measures: coherence, which measures how well the terms in a topic are grouped in a document, and exclusivity, which captures how well topics split documents into different groups.*

How these metrics (coherence and exclusivity) change according to different k, can be found in Supplementary material 5.

*Considering the highest possible coherence and the highest possible exclusivity, we chose k = 15.*

Lines 225 - 229. Can the authors explain the magnitude of drought terminology a little here and how this fits into their work (i.e., major drought vs others)? I think this would be helpful to remind the reader of. Also, it would be interesting if the authors could note any patterns in the length of the official drought and the number of articles during that same period, maybe in a table format. It's a little hard to tease that out from Figure 3.

First part of this comment is not valid anymore in our revised manuscript (we do not use the bubble plots for chi-square test). For the second part of the comment about visualization, we offered a new figure based on the best-fitting combination of variables according to negative binomial regression.

Line 250. I'm a little confused about how the topic labels were developed. Is this a result/output from the model or is this something the authors created. Also what is done for keywords that overlap many topic areas (e.g., "drought" is showing up as a keyword in all topics). I think more detail of the procedures used is needed to make this more clear.

Apologize for this. We clarified this point in L 210. Naming the topics after the topic modeling is a common procedure.

*After reviewing the keywords, the authors assigned labels that best describe the topics.*

It is also common for some keywords to appear across different topics, as LDA is based on a mixed membership model.

Line 314. See my comment for lines 225-229. That information could be of interest to this discussion point.

The lines 225-229 in old manuscript are no longer available in the revised manuscript (we do not use the bubble plots for chi-square test). Thank you for this comment.

Lines 331-339. I like that the authors include this detail on the flipside case and give regional context/specifics on why it might not have shown up in the news.

This point is now addressed in our new discussion about limitations, i.e., 6th and 7th paragraphs in Section 4.1, that described of fixed temporal windows and spatial granularity.

Lines 380-385. Can the authors also discussion some of the studies that mentioned in the introduction here? I think that comparison would be helpful to put in the context of these results other similar studies. I'm thinking the studies that are mentioned around lines 58-70.

While it is really a good point to make an extendable discussion, we found it a bit challenging for the earlier works to focus on different events of different countries. Instead, we explained the possibility to overcome the abovementioned limitations about our corpus, by considering non-newspaper documents, as we described at the end of Chapter 4.2.

*Extending the corpus to incorporate more genres of writing about drought, e.g., by incorporating resources including the UK Drought Inventory (e.g., Parsons et al., 2019), and the European Drought Impact Inventory (EDII) (e.g., Stahl et al., 2016), as well as social media could provide valuable additional perspectives.*

Lines 385-390. I recommend moving this text to the methods section where this method is discussed.

We removed this part. Thank you.

Lines. 402 – 404. "In other words, the underrepresentation of media coverage at the stage of hydrological droughts can result in the lack of public awareness for 'creeping' droughts,…" Can the authors explain how their study supports this statement? It is a logic jump for me and I'd like to see more specific examples from the study results given.

The conclusion was completely rewritten to reflect our revised analytic methods and results.

Line 405. Can the authors make the connection to your results a little clearer here? Is there a specific policy priority example they might recommend given the results of this study?

While loosely connected, we mentioned in the Introduction (7th paragraph) of the UK government's policy to deal with climate change.

Table 1. What does "frex estimation" mean? I recommend the authors including "(org)" in the topic labels section so it would read "ten frequent organization entities in descending order of frequency (org)".

We described the frex keyword in the manuscript:

*…frex scoring algorithm (i.e., keyword extraction methods considering both frequent in and exclusive to a topic of interest)* (LL208-209)

The column name for keywords is now phrased as follows:

*Probability-based keyword (prob), frex keywords (frex), and frequent organization names (org). In descending order of frequency*

Figures 1 and 2. Can the authors give some indication in the image or caption what the color scale refers to on these plots? This information is not currently explained.

These figures are now deleted after the major change in methods.

Figure 3. How do the authors define "drought-prone condition" in this figure? This isn't clear to me.

We explained this in the followings (LL257-258):

*The plot also showed that most newspaper reporting for major droughts coincided with the drought-prone conditions, i.e. 12-month CET anomalies > 0, 3-month SPI < 0, and 1-month SGI < 0 (see the yellow shades).*

Figure 4. This is an interesting figure but there are some aspects that I'm confused by. I'm not seeing subsets 1-3. I suggest the authors write out what the y axis variables mean in the caption or give a key to connect them such as "6-month average temperature anomaly (temp.6mo.avg)". I suggest the authors use a similar color scale and shape for the three variables. For example, looks like the article count is a square vs the other two variables, which are circles. Can they also provide the color scale? The text in this figure is a bit small, but the authors should be able to work with HESS to make sure the proof text is landscape oriented (vs portrait oriented).

According to our major change in method, we arranged the variables in this figure to follow the best-fitting variables: CET12, SPI3, SGI1. The names of the variables are put together on the left side.

Figure 5. This figure is also interesting but can the authors provide more information about how to interpret the results. Does the size of the color square indicate it's importance in the tree? What do the N/A values mean and can they be combined (or removed from the analysis and plot)? The text in this figure is a bit small, but the authors should be able to work with HESS to make sure the proof text is landscape oriented (vs portrait oriented).

For the case of using unsupervised text analytics, i.e., topic modeling, it is common to find such irrelevant topics due to the variety of topics in the text corpus. Such cases, we did not name the topics, and left them as N/A.

The size of colored square indicates the size of newspaper articles. For this figure (now captioned as Figure 3), we replaced the texts with just numbers in the figure, and added a legend where readers can find the corresponding labels to the numbers.

Supplemental figures. Can the authors provide captions for these? I'm not seeing those.

Thanks for your comment. We added captions for the figures.

Supplemental doc pg 1 and 2. Can the authors give some indication in the image or caption what the color scale refers to on these plots? This information is not currently explained.

In our revised manuscript, we dropped the chi-square analysis to replace the statistical analysis with binomial regression analysis. According to the new analysis, you can find the relevant materials in the Supplementary Materials from 2-4.

Figure S4b (I think?). The authors can use set the y-scale to be free parameter (scales = "free_y") in the facet wrap in R. That way they'll be able to show years with fewer counts on their own scale (rather than the scale 0 to 200).

We agree that this is a good way to make sure the bar chart is visible. Nonetheless, we fixed the y-scale so that the trend of media volume for the last 24 years can be clearly captured. For your information, this figure is now located in Supplementary Material 7.

**RC3.**

**Notification on major change**

We would like to thank all three reviewers and the editor for taking the time to read and provide thoughtful comments on our manuscript.

Before addressing the detailed comments line by line, we would like to outline the major changes made in our revised manuscript. These changes address shared concerns raised by more than one reviewers and we assured that these will enhance the overall integrity and robustness of our work.

In the first part of our analysis, we aimed to examine the relationship between explainable variables and the volume of news media attention. Originally, this analysis involved precipitation and temperature anomalies, assessed alongside a chi-square test to explore seasonal variations. In the revised manuscript, we replaced the precipitation anomaly with WMO drought indices (SPI/SSI) and incorporated additional variables such as SGI and seasonality. Consequently, we transitioned from using chi-square analysis to negative binomial regression analysis. This methodological change allowed us not only to identify the best-fitting combinations of variables in explaining the volume of newspaper articles but also to quantify the contribution of each variable.

The second part of our analysis applied a multi-scalar text analysis approach, combining computational topic modeling with qualitative close reading. While the methodology in this section largely remains unchanged, we increased the number of topics from 10 to 15, based on coherence and exclusivity tests. This adjustment enabled us to uncover rarer but meaningful topics within the corpus, which further enriched our findings. For example, this revision illuminated the role of seasonality in influencing media attention, bridging insights from the statistical analysis.

These changes would make some comments from the reviewers no longer applicable. Below, we present detailed responses to the comments from the three reviewers. Thank you for your time and consideration.

**Overall comments**

1. The discussion about the results displayed in the figures is lacking, such that the reader may have a hard time understanding the take home or some of the data. I would recommend additional content within the manuscript to walk the reader through the results.

Thank you for the comment. In the revised manuscript, we updated the results in line with the **'Major change'** and revised the discussion section, placing special emphasis on expanding the scientific implications.

2. Many of the figures are challenging to read and the text overlaps. Further, ensure that all figures satisfy any color-blind considerations.

Thank you for the comment. We believe the comment about text overlaps is referring to Figure 3. We replaced the texts with just numbers in the figure, and added a legend where readers can find the corresponding texts to the numbers. In addition, we also applied the colorblind palette, 'safe' from

rcartocolor (Nowosad, 2018). We also tested how effective this new palette is, and the test result is as below.

[Figure]

Additionally, we modified the color palette of Figure 1 and 2 to abide by colorblind friendly one as well (Wong, 2011).

3. Some of the paragraphs in the discussion have a really clear take home lessons/message within the paragraph and making them explicit (see the first in 4.1). However, in many of the paragraphs, the phrasing/format is a bit awkward. See Lines 321-325 as an example. Much of the content in the discussion seems disjointed and doesn't have the coherence needed to unpack this complex research. As such, I'm having a hard time following and understanding the primary points the authors are trying to make.

Thank you for the comment. In the revised manuscript, we made a thorough revision to summarise key take-home message, and to make the discussion consistent and coherent to such messages.

4. I would recommend the authors to do a deeper dive in the existing literature to further contextualize their results to current knowledge beyond the work by Marsh and Dayrell in the Discussion. The big "so whats" seem to be missing. For example, studies have been done that have looked at changes in water consumption based on media coverage of extreme events: https://www.science.org/doi/10.1126/sciadv.1700784. See other examples, which may be more or less pertinent since the geographies are different (certainly no obligation to include, but added to encourage further analysis, reflection, and contextualization of the within this article): Molina et al. https://www.science.org/doi/10.1126/sciadv.1700784, Rutledge-Prior and Beggs https://onlinelibrary.wiley.com/doi/abs/10.1111/ajph.12759, Helm Smith et al. https://journals.ametsoc.org/view/journals/bams/101/10/bamsD190342.xml. As such, I encourage the authors to ensure that the novelty and enhancement to the field is obvious and that there is a clear, explicit link between the results detailed in Section 3 and the discussion of said results. There are limited call-backs and I think that diminishes the potential impact of this piece.

Thank you for the comment. In our revised manuscript, we clarified our research motivation in 3rd paragraph of introduction – why it matters to investigate the relationship between media attention and drought events, so that we can better contextualize the paper.

Also, with our major change in methodologies, we focused on eliciting in-depth discussion, together with the clarification of major takeaway (i.e., for shorter version, please refer to Abstract; for full-length insights, please refer to Discussion).

Still, we reviewed the literature works you recommended:

One paper (Quesnel & Ajami, 2017: https://www.science.org/doi/10.1126/sciadv.1700784) has been already included in our old/new version of manuscript.

Vargas Molina et al. (https://www.tandfonline.com/doi/full/10.1080/17477891.2021.1932712), was a good example to feature the usefulness of newspaper analysis of droughts in understanding its narratives over time. We took this work to enrich our Introduction, but it a bit difficult to blend in our discussion, for its semi-qualitative analysis as well as study site (Spain and Brazil).

We also reviewed Rutledge-Prior and Beggs (2021) (https://onlinelibrary.wiley.com/doi/abs/10.1111/ajph.12759), which explored the media narratives in Australia. While this work was interesting to find the media narratives in Australia to heavily focus on

agricultural impacts, we found this point somewhat overlap with our discussion points with Stahl et al. (2016), which elaborated on the comparisons in European scope.

5. I am still a little unclear on the link between the demand side and its effect on drought and how that's drawn out in this paper. As mentioned in the limitations section, linking water demand (population density, industrial needs) to drought coverage could strengthen the analysis and offer a more comprehensive understanding of media biases.

Thank you for point this out. The demand side of water resources is not directly presented in this work, but we can only take a glimpse of the importance of such topic in newspaper narratives through the topics related to restrictions on water use. To clarify this point, we made a substantial change in Introduction.

Still, we find this aspect of water demand interesting. As such, we attempted to get finer in spatial analysis by taking regional indices and examine whether topics related to demand are more prevalent in areas with higher urban demand. However, we were unable to achieve this, due to the nature of our data source: (1) the number of newspaper articles per month was insufficient to analyze regional variations, and (2) assigning geographic entities to newspaper articles posed a significant technical challenge, requiring substantial manual effort. We have clarified this limitation at the end of Chapter 4.2.

6. The national aggregation of meteorological variables may obscure regional variations in drought severity and media coverage. Incorporating more granular data could provide further insights into regional biases or differences in public perception. If not included, a further discussion about the results may be warranted.

We understand that your earlier comment addressed the spatial granularity of data, and this comment pertains to the spatial granularity of drought indices. We found this observation relevant and tried to account for regional variations in drought indices, as detailed below. Our initial assumption was that the London metropolitan area (i.e., SE in the legend) might exert a stronger influence on media attention due to relatively drier conditions and higher demand. However, we concluded that there is no strong evidence of a consistent 'drier condition' in the London metropolitan area. We have clarified this as a limitation, which is discussed at the end of Chapter 4.1.

[Figure]

7. Some topics (particularly the N/As) may need further detail in their relevance to the central theme of droughts. Clarifying could improve the interpretability of results. Further, it's the ordering of Table 1 is confusing when it's not numerical.

Thank you for the comment. As part of extending our topic modeling, we updated Table 1. Specifically, we revised the probability-based keywords, FREX-estimation keywords, and frequent organization names compared to the previous version of the manuscript. With the increase in the number of topics from 10 to 15, we identified three topics that were not relevant to our focus and provided an explanation for why these were excluded. It is worth noting that in unsupervised text analytics, such as topic modeling, it is common to encounter irrelevant topics due to the diverse range of subjects present in the text corpus.

Lastly, we reordered the topics in descending order of their overall probability within the corpus to improve clarity for readers.

8. The authors did a good job to ensure that they attribute and discuss the appropriateness of your methods through citations.

Thank you. We appreciate your feedback!

---

## Referee Report (RR2)

HESS Review for egusphere-2024-1844

**General Comments:**

After reading through the authors responses, they put in a lot of effort to address the feedback from all three reviewers. They addressed all my comments from the first round. Furthermore, their new statistical analysis combined with drought indices adds more nuance to the study's original findings, which is a great to see.

I had one lingering general comment that is described in more detail in my line comments below. Briefly, I'd like the authors to add more discussion on whether/how the interpretation of their results may differ if they are to choose one of the more parsimonious models in Table S1.

Based on my general comment and the handful of line edits below, I recommend this manuscript be accepted with minor revisions. Thanks again to the authors for their interdisciplinary approach to looking at drought through the lens of newspaper media. I found the paper very interesting and am grateful for the opportunity to serve as a reviewer.

**Line Edits**

Line 139. I suggest removing the word "package" as, from my reading, the authors are referring to the entirety of the R software, not a specific R package. The final sentence would then read: "The process was performed in R, version 4.3.2..."

Lines 153-159. Pointing out that this text is all underlined because I'm not sure if that was intentional. I do not think the underline is needed.

Line 178-179. Can the authors please provide a sentence explaining what R package they used to do the negative binomial regression analysis and cite the package? I believe it's, the MASS R package, correct?

Line 180. If AIC values are within two units of one another, they are not (statistically) significantly different. As a result, it's typical to choose the most parsimonious of the models tested within the two-unit difference. That said, I recommend the authors consider whether/how the findings of their work may differ if they were to use, for example, the model with only SPI12 and seasonality (Table S1 row 3). I'd like to see more discussion of this in the results section. It would also be great to include a third line for that model on Figure S2, if that helps to show the comparison in performance.

Lines 383-386. I think this point is very interesting and appreciate the authors adding this discussion of nuance here and in the abstract.

Table 1. I appreciate the authors including this table with the R model outputs and their interpretations, but I recommend they move this level of detail to the supplement.

Figure 2. Overall, this is an interesting visual! I wanted to point out that I do not see a lilac-colored grid on the top; it looks grey in the attached PDF. Since there is no color scale bar given, I recommend the author's explicitly give a description in the caption that lighter hue saturation indicate higher article counts and hydroclimatic anomalies (either larger negative or positive) whereas darker hue saturation indicate the opposite.

Table S1. Is there a way for the authors to include R squared or something similar in this table that would give a sense of how much variation in article counts each model explains?

---

## Author Response (AR2)

**Response to two reviewers' comments (2nd round)**

**Droughts and Media: when and what do the newspapers talk about the droughts in England?**

**Comments from the Editor**

One of the original reviewers was available and they have reviewed the manuscript based on the comments of all the 3 original reviewers. The authors indeed put in lots efforts to address the comments and improve the manuscript. And I also thank authors for their open and welcoming attitude towards the reviews.

In this round of review, a new reviewer was also invited from industry background, who raised a number of questions for authors to think about. I would like to invite the authors to address these comments, especially in the discussion of their paper. This can give authors an opportunity to see how a practitioner view/read their paper.

We would like to thank both the editor and two reviewers for the comments on our paper. We are happy to see that our previous revisions have been appreciated and have now further revised the manuscript as requested. These minor revisions on the one hand address the specific suggestions made by Referee #2 and on the other attempt to clarify some of the points from the new referee (Referee #4) with an industry background. Additionally, we have made minor changes only to an extent to improve the readability of the manuscript.

**Referee #2**

General Comments:

After reading through the authors responses, they put in a lot of effort to address the feedback from all three reviewers. They addressed all my comments from the first round. Furthermore, their new statistical analysis combined with drought indices adds more nuance to the study's original findings, which is a great to see.

Thanks a lot for taking the time to look at our paper again and for your positive and constructive feedback. We are really pleased that you like the increased nuance of the paper's findings.

I had one lingering general comment that is described in more detail in my line comments below. Briefly, I'd like the authors to add more discussion on whether/how the interpretation of their results may differ if they are to choose one of the more parsimonious models in Table S1.

We address this point, and the others, in the specific line edits.

Based on my general comment and the handful of line edits below, I recommend this manuscript be accepted with minor revisions. Thanks again to the authors for their interdisciplinary approach to looking at drought through the lens of newspaper media. I found the paper very interesting and am grateful for the opportunity to serve as a reviewer.

Many thanks.

Line Edits

Line 139. I suggest removing the word "package" as, from my reading, the authors are referring to the entirety of the R software, not a specific R package. The final sentence would then read: "The process was performed in R, version 4.3.2…"

Thanks, you are right, we replaced this as "R Studio version 4.3.2".

Lines 153-159. Pointing out that this text is all underlined because I'm not sure if that was intentional. I do not think the underline is needed.

This underlining was to highlight some changes, we have removed it now. Thanks for pointing it out.

Line 178-179. Can the authors please provide a sentence explaining what R package they used to do the negative binomial regression analysis and cite the package? I believe it's, the MASS R package, correct?

Correct. We have specified this in the paper and added a reference to the package ("…find the best-fitting model using the MASS package in R (Venables & Ripley, 2002)."). This also addresses a point made by Reviewer #4 (see below).

Line 180. If AIC values are within two units of one another, they are not (statistically) significantly different. As a result, it's typical to choose the most parsimonious of the models tested within the two-unit difference. That said, I recommend the authors consider whether/how the findings of their work may differ if they were to use, for example, the model with only SPI12 and seasonality (Table S1 row 3). I'd like to see more discussion of this in the results section. It would also be great to include a third line for that model on Figure S2, if that helps to show the comparison in performance.

Thanks for pointing this out. We have added plots of two of the more parsimonious models to Figure S2 and discussed these results briefly in the results.

[Figure]

**CET3, SPI3, SGI1, Seasonality, Lag**

[Figure]

**SPI12, Seasonality, Lag**

[Figure]

**Figure S1 Comparisons between actual article counts (blue) and modelled article counts (red) for three best models**

In this study, the best model with the smallest AIC was to use CET-12, SPI-3, SG-I, and Seasonality, and the lag effect of the media (top), the second-best model to use CET-3, SPI-3, SGI-1, Seasonality, and the lag effect (middle), and the third best model, and the most parsimonious one, to use SPI-12, Seasonality, and the lag effect (bottom). Since their AICs differ by only a couple digits (please refer to Table S1), we concluded that it is reasonable to explain the most parsimonious model (bottom), as well as the best model (top). As a result, we elaborated on this point with an updated Table 1.

**Table 1** Statistical results of negative binomial regression models: the best AIC model and the parsimonious model ($p$-values: '***' < 0.001, '**' < 0.01, '*' < 0.05, '.' < 0.1)

| Variables | The best AIC model | | | Parsimonious, yet 3rd best model | | |
|---|---|---|---|---|---|---|
| | Best-fitting variable | | Negative binomial coefficient (p-values) | Best-fitting variable | | Negative binomial coefficient (p-values) |
| Temperature | CET-12 | | 0.395 (.) | - | | - |
| Precipitation | SPI-3 | | -0.415 (***) | SPI-12 | | -0.835 (***) |
| Groundwater | SGI-1 | | -0.868 (***) | - | | - |
| Seasonality (Autumn as baseline) | To include | Winter | 0.693 (*) | To include | Winter | 0.670 (*) |
| | | Spring | 1.065 (***) | | Spring | 1.107 (***) |
| | | Summer | 1.108 (***) | | Summer | 1.206 (***) |
| Lagging effect of media attention | To include | | 0.046 (***) | To include | | 0.050 (***) |
| Intercept | | | -0.736 (**) | | | -0.528 (*) |

Lines 383-386. I think this point is very interesting and appreciate the authors adding this discussion of nuance here and in the abstract.

Thanks for this - we're glad that this helped.

Table 1. I appreciate the authors including this table with the R model outputs and their interpretations, but I recommend they move this level of detail to the supplement.

We agree to this point. We relocated the descriptive notes about the results, including coefficients and statistical significance, in the manuscript. In addition, according to your suggestion, we looked into the parsimonious model (i.e., the model with the third lowest AIC), which allowed us to make a comparison about negative binomial coefficients between the best model and the parsimonious model.

Figure 2. Overall, this is an interesting visual! I wanted to point out that I do not see a lilaccolored grid on the top; it looks grey in the attached PDF. Since there is no color scale bar given, I recommend the author's explicitely give a description in the caption that lighter hue saturation indicate higher article counts and hydroclimatic anomalies (either larger negative or positive) whereas darker hue saturation indicate the opposite.

Thanks for picking this up - the grid is in fact now in grayscale. We have modified the caption to refer to grayscale and indicate that saturated hues refer to drought-prone anomalies.

Table S1. Is there a way for the authors to include R squared or something similar in this table that would give a sense of how much variation in article counts each model explains?

We use the glm function from the MASS package from R, which by design doesn't output (pseudo) $r^2$ values. We took up your other suggestions, and plotted the modelled distributions in Figure S2 as you suggested above.

**Referee #4**

Explaining the statistics to choose the binomial distribution. The R code shows that the author did run tests to determine the distribution and statistically should be explained for scientific validity.

We are working with count data, which are not normally distributed. Therefore we used a negative binomial model as discussed in the methods (L165), which is appropriate to non-normal, highly skewed data.

The research does draw into question the relevance of this issue, as the media uptake of drought conditions should reflect the authority's response. Regardless of what words are used to find what articles are relevant to how reported droughts are aligned with the scientific community, the accuracy of when they are reported, if not completely false or not completely misaligned by time period or scientifically accurate, is not a major determining factor of what the response to droughts should be.

The aim of our paper was to explore how the media report droughts. We don't claim that media reporting **should** influence the response to droughts. As we explained in the introduction (L50) media reporting "influences public perception and societal responses" (e.g. to restrictions) which in turn influence the effectiveness of policy responses. We have edited the text in the introduction to make this point clearer.

Social media is an unaddressed component of news during 2022 and was not addressed extensively. This is a main difference between the reporting between the main years chosen with significant droughts. This is a limitation for the data explored in 2022. Has this been catered for in the code for the binomial regression for the 2022 drought?

Although social media have been the subject of great interest, our paper uses newspaper reporting as a source. As we explained in the discussion, using only a single genre of reporting is a limitation and we recognise that including "social media could provide valuable additional perspectives" (L464).

This paper may be deemed a media analysis, not of scientific literature. The climactic factors during a particular time period are important. However, in terms of when the droughts are reported in the media, this has not been referenced within the scientific literature. Are there regulatory obligations? How should this be applied to current events?

We are not quite sure what is meant here, but it is indeed the case that one could study the genre of scientific papers as a different source. In this study, we focused on newspapers as a key arena for mass communication, which reflects the bi-directional relationships both from the government/authority and the public.

With respect to regulatory obligations about the reporting of droughts in the UK, there are none as far as we are aware, since the UK has a free press with very limited restrictions on reporting except with respect to national security issues (Wilkinson, 2009).

Wilkinson, N. J. (2009). *Secrecy and the media: the official history of the United Kingdom's D-notice system.* Routledge.